# Merge to Remember: Sharpness-Aware Isotropic Merging for Continual Learning

Qun Yang [1]  Enneng Yang [2 3]  Wei Chen [1]  Li Shen [2 3]  Long Lan [1]

## Abstract

Continual learning with large pre-trained models offers significant potential for cross-task knowledge accumulation, but faces critical challenges such as catastrophic forgetting and parameter interference, especially when historical data is unavailable. Existing approaches typically rely on sequential fine-tuning or model merging strategies, yet often overlook the impact of loss landscape sharpness and dominant singular value directions, which leads to subspace misalignment and severe knowledge forgetting. In this paper, we propose the Sharpness-Aware Isotropic Merging (SAIM) framework, which introduces targeted optimizations in both the fine-tuning and merging stages to address these issues. Specifically, SAIM consists of two synergistic modules: (1) a Sharpness-Aware Block Coordinate Descent (SA-BCD) optimizer that guides the model toward flatter minima and selectively updates the most task-sensitive parameters, thereby mitigating parameter interference and enhancing robustness; (2) an adaptive isotropic merging algorithm that dynamically balances the singular value spectrum across tasks, effectively preventing the model from overemphasizing any single task direction, maintaining balanced knowledge representation, and improving subspace alignment. Extensive experiments on vision and language benchmarks demonstrate that SAIM achieves 5-10% higher accuracy than existing methods and maintains robust performance as the number of tasks increases. Our code are available at https://github.com/Yangqun123456/SAIM.

[1]College of Computer Science and Technology, National University of Defense Technology [2]School of Cyber Science and Technology, Shenzhen Campus of Sun Yat-sen University [3]Guangdong Laboratory of Artificial Intelligence and Digital Economy (SZ). Correspondence to: Long Lan <long.lan@nudt.edu.cn>.

*Proceedings of the 43rd International Conference on Machine Learning*, Seoul, South Korea. PMLR 306, 2026. Copyright 2026 by the author(s).

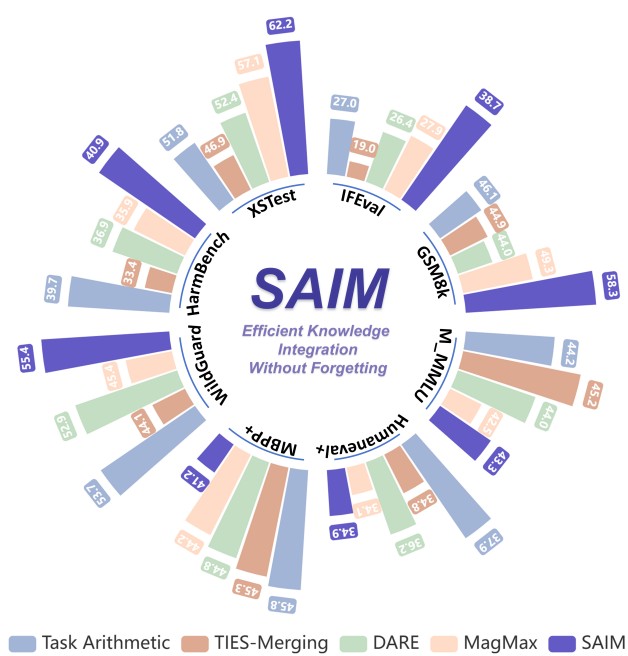

*Figure 1.* Performance comparison of SAIM against baseline methods on MergeBench using *Llama-3.2-3B*.

## 1. Introduction

With the rapid development of deep learning, large pre-trained models have become the cornerstone of numerous artificial intelligence applications (Radford et al., 2021; Dosovitskiy et al., 2020). However, when these models are required to continually learn from new tasks, they inherently face two fundamental macroscopic challenges: catastrophic forgetting and generalization degradation (Kirkpatrick et al., 2017; Lopez-Paz & Ranzato, 2017). These challenges are fundamentally driven by parameter interference, where conflicting gradient updates across tasks misalign shared parameters, leading to structural representation distortion. Traditional continual learning methods typically adopt sequential fine-tuning; while this enables adaptation to new tasks, its sequential nature biases the model toward recent data and erodes earlier knowledge (Buzzega et al., 2020; Zenke et al., 2017). Additionally, existing strategies such as replay buffers, regularization, or parameter expansion often struggle to balance efficiency and effectiveness, frequently

failing to address data privacy and storage concerns (Rusu et al., 2016; Yoon et al., 2017; Farajtabar et al., 2020).

In recent years, model merging techniques have shown great potential in continual learning scenarios, particularly excelling in applications with data privacy protection and storage constraints (Ilharco et al., 2022; Wang et al., 2024; Li et al., 2023). Task arithmetic (Ilharco et al., 2022) constructs multi-task models, MagMAX (Marczak et al., 2024) reduces parameter interference via maximum magnitude selection, orthogonal projection-based continual merging (Tang et al., 2025b) further minimizes task interference through sequential projections, while mixture-of-experts approaches (Qiu et al., 2024) leverage dynamic gating and low-rank adaptation for robust continual integration. However, when applied to continual learning scenarios, these methods face two main challenges: on one hand, they typically require maintaining the cumulative results of all historical task vectors, increasing storage overhead (Izmailov et al., 2018; Wortsman et al., 2022); on the other hand, these methods have limitations in addressing parameter update conflicts between different tasks, resulting in severe interference issues and representation space distortion. More importantly, the fine-tuning and merging processes are often treated as independent stages, overlooking their potential synergy, which leads to loss landscape incompatibility and subspace misalignment, ultimately affecting knowledge retention and transfer efficiency.

To address these challenges, we propose a novel continual learning method—Sharpness-Aware Isotropic Merging (SAIM)—which unifies the fine-tuning and merging processes into a synergistic framework to achieve efficient continual knowledge acquisition and retention. SAIM is based on two key technical innovations:

First, *in the fine-tuning stage*, we propose the Sharpness-Aware Block Coordinate Descent (SA-BCD). SA-BCD introduces a critical perspective: treating the potential parameter drift induced by future model merging as a **predictable perturbation**. Conventional fine-tuning often converges to sharp minima where slight weight shifts (caused by merging) lead to drastic performance drops. By explicitly optimizing for flatness, SA-BCD ensures the learned solution resides in a high-volume basin that remains valid even after the weights are shifted.

Second, *in the merging stage*, we introduce an adaptive isotropic merging algorithm to resolve inter-task parameter interference. By dynamically rebalancing the singular value spectrum, this method mitigates the spectral bias where recent or dominant tasks overshadow others. This isotropic constraint ensures an equitable utilization of the parameter space and facilitates superior subspace alignment across tasks, effectively preserving the structural robustness established during fine-tuning. The effectiveness of our approach is illustrated in Figure 1, where SAIM consistently outperforms existing methods in both knowledge retention and adaptation in most tasks.

Our main contributions are as follows: ❶ We treat fine-tuning and merging as synergistic processes, where flat solutions facilitate subsequent parameter integration and merging provides a better starting point for new tasks. This unified framework enhances knowledge retention and improves overall continual learning performance. ❷ We propose a novel approach that jointly optimizes fine-tuning and merging to mitigate catastrophic forgetting and parameter interference. By adaptively balancing the singular value spectrum, our method enhances subspace alignment across tasks, alleviates forgetting, and eliminates the need to store historical task vectors. ❸ We systematically evaluate SAIM across three vision architectures and three language model architectures, as well as varying numbers of tasks, demonstrating its effectiveness in reducing catastrophic forgetting and improving adaptation to new tasks. Furthermore, the SA-BCD can be combined with most existing model merging approaches to further enhance performance.

## 2. Preliminaries

Before presenting our method, we briefly review the fundamental problem setup of continual learning and key theoretical foundations.

### 2.1. Problem Setup

In the continual learning (CL) setting, we focus on the following problem: Given a large pre-trained model with parameters $\theta_0$, and a sequence of $N$ downstream tasks $\mathcal{T} = \{1, 2, \ldots, N\}$ arriving sequentially, each task $t$ is associated with an independent dataset $D_t = \{(x_j^{(t)}, y_j^{(t)})\}_{j=1}^{n_t}$, where $x_j^{(t)} \in \mathcal{X}$ and $y_j^{(t)} \in \mathcal{C}_t \subset \mathcal{Y}$, with $\mathcal{C}_t$ being the label space for task $t$. Our goal is to efficiently integrate the newly obtained task expert $\theta_t$ with the current model parameters $\theta_{t-1}^{merged}$ at each step, and finally obtain a unified model compatible with all seen tasks.

As in conventional continual learning, we assume that no historical training data is accessible during the model integration process; all knowledge transfer and integration occur in parameter space. The continual merging process can be formulated as:

$$\theta_t^{merged} = \text{ContinualMerge}(\theta_{t-1}^{merged}, \theta_0, \theta_t) \quad (1)$$

where $\theta_0^{merged} = \theta_0$. The final objective is to minimize the average loss over all seen tasks:

$$\min_{\theta_N^{merged}} \frac{1}{T} \sum_{t=1}^{T} \mathbb{E}_{(x,y)\sim\mathcal{D}_t} \mathcal{L}(h_t(\mathcal{F}(x; \theta_N^{merged})), y) \quad (2)$$

where $\mathcal{F}(x; \theta)$ denotes the backbone network, $h_t$ is the task-specific head, and $\mathcal{L}$ is the loss function.

## 2.2. Sharpness-Aware Minimization and Flatness

In the process of fine-tuning and merging deep models, the "flatness" of the local minima where model parameters reside is closely related to generalization. Flat minima are more robust to parameter perturbations, helping the model accommodate knowledge from multiple tasks and reducing catastrophic forgetting.

Sharpness-Aware Minimization (SAM, Foret et al., 2021) optimizes parameter flatness by simultaneously minimizing the loss value and its sharpness. The optimization objective is:

$$\min_{\theta} \left[ \max_{\|\epsilon\|_2 \le \rho} \mathcal{L}(\theta + \epsilon; \mathcal{D}) - \mathcal{L}(\theta; \mathcal{D}) \right] + \mathcal{L}(\theta; \mathcal{D}) \quad (3)$$

where $\epsilon$ is a perturbation vector and $\rho$ controls the radius of the neighborhood. For efficiency, SAM uses a Taylor approximation to simplify the objective as:

$$\min_{\theta} \mathcal{L}(\theta + \hat{\epsilon}; \mathcal{D}), \quad \text{where} \quad \hat{\epsilon} = \rho \frac{\nabla_{\theta} \mathcal{L}(\theta; \mathcal{D})}{\|\nabla_{\theta} \mathcal{L}(\theta; \mathcal{D})\|} \quad (4)$$

By optimizing with SAM, the model is more likely to converge to flat minima, improving robustness and multi-task compatibility.

## 2.3. Task Arithmetic and Subspace Alignment

Task Arithmetic (TA, Ilharco et al., 2022) provides a framework for constructing multi-task models. The knowledge of each task $t$ can be represented as a task vector $\tau_t = \theta_t - \theta_0$. By weighted summation of these task vectors, multiple task models can be merged:

$$\theta_{merged} = \theta_0 + \sum_{t=1}^{T} \alpha_t \tau_t \quad (5)$$

where $\alpha_t$ denotes the task weight. However, as the number of tasks increases, simple vector addition can lead to parameter conflicts and interference (Tang et al., 2025b).

Recent studies have found that merging performance depends on the degree of alignment among the subspaces spanned by the task vectors (Gargiulo et al., 2025). The Subspace Alignment Ratio (SAR) measures the overlap between the merged model and task-specific parameter updates:

$$\text{SAR}(\Delta_{\text{src}}, \Delta_{\text{trg}}; k) = \frac{\|\pi_{k,\text{trg}} \Delta_{\text{src}}\|_F}{\|\Delta_{\text{src}}\|_F} \quad (6)$$

where $\Delta_{\text{src}}$ and $\Delta_{\text{trg}}$ denote the source and target task update matrices, respectively, and $k$ represents the number of

dominant left singular vectors used to compute the subspace alignment ratio. $\pi_{k,\text{trg}}$ is the projection operator onto the $k$-dimensional principal subspace of the target update matrix. A higher SAR value implies that the task vector retains its essential directionality within the merged subspace. In continual learning, maintaining high SAR is critical, as it ensures that new updates do not destructively interfere with the geometric structures encoding previous knowledge.

## 3. Approach

Continual learning traditionally relies on sequential fine-tuning, using regularization (e.g., EWC (Kirkpatrick et al., 2017)) or experience replay (Buzzega et al., 2020) to mitigate catastrophic forgetting. Recently, model merging methods such as MagMAX (Marczak et al., 2024) and OPCM (Tang et al., 2025b) have enabled continual learning by integrating knowledge from multiple task-specific models, thereby further reducing forgetting. However, most approaches treat fine-tuning and merging as separate processes. In this work, we explicitly consider both sharpness-aware fine-tuning (Sec. 3.1) and self-adaptive isotropic merging (Sec. 3.2), and demonstrate that jointly optimizing these two stages leads to more effective knowledge integration and retention in continual learning.

### 3.1. Sharpness-Aware Block Coordinate Descent Optimizer

To address parameter interference and catastrophic forgetting, we propose the **Sharpness-Aware Block Coordinate Descent (SA-BCD)** optimizer. In the paradigm of continual learning via model merging, the integration of a new task model essentially constitutes a **parameter perturbation** to the current model state. SA-BCD is explicitly designed to preemptively address this challenge by modeling and adapting to worst-case parameter perturbations during the fine-tuning stage. By forcing the model to accommodate these simulated disturbances, SA-BCD guides convergence towards regions that are not only flat but intrinsically robust to parameter shifts. This process effectively prepares the model for the inevitable parameter drift caused by subsequent merging operations, ensuring that the model retains its original capabilities even after the integration of future tasks.

Concretely, at each iteration, SA-BCD computes the gradient of the current parameters $\Theta_t$ with respect to the loss function $\mathcal{L}(\Theta_t, \mathcal{D})$, and updates the first-order momentum as follows:

$$m_t = \beta_1 m_{t-1} + (1 - \beta_1) \nabla_{\Theta} \mathcal{L}(\Theta_t, \mathcal{D}), \quad (7)$$

where $\beta_1$ is the first-order momentum coefficient ($\beta_1 = 0.9$ by default). The magnitude of $|m_t|$ is then used to select the top $p\%$ of parameters with the largest absolute values,

denoted as $\Omega_t = \text{Top}_p(|m_t|)$ ($p = 30\%$ by default). We exclusively update these parameters to ensure efficient adaptation to the current task, while freezing others to preserve the model's existing capabilities. For the selected subset $\Omega_t$, SA-BCD introduces a sharpness-aware perturbation mechanism to explore the worst-case direction in the loss landscape:

$$\epsilon_{t,\Omega}^* = \rho \frac{\nabla_\Theta \mathcal{L}(\Theta_t, \mathcal{D})_{\Omega_t}}{\|\nabla_\Theta \mathcal{L}(\Theta_t, \mathcal{D})_{\Omega_t}\|_2} \tag{8}$$

where $\rho$ is a hyperparameter controlling the perturbation magnitude, and $\nabla_\Theta \mathcal{L}(\Theta_t, \mathcal{D})_{\Omega_t}$ denotes the gradient restricted to the subset $\Omega_t$. This perturbation enables the optimizer to probe the geometry of the loss surface, steering the model away from sharp minima and towards flatter, more generalizable regions. Next, the optimizer updates the second-order momentum and applies bias correction as in Adam (Kingma & Ba, 2014): $v_t = \beta_2 v_{t-1} + (1 - \beta_2)(\nabla_\Theta \mathcal{L}(\Theta_t, \mathcal{D}))^2$, $\hat{m}_t = m_t/(1 - \beta_1^t)$, $\hat{v}_t = v_t/(1 - \beta_2^t)$. The gradient is then recomputed at the perturbed point $\Theta_t + \epsilon_{t,\Omega}^*$, yielding $g_t' = \nabla_\Theta \mathcal{L}(\Theta_t + \epsilon_{t,\Omega}^*, \mathcal{D})$. Finally, SA-BCD updates only the parameters in $\Omega_t$, while keeping the others unchanged. The update rule is:

$$\Theta_{t+1,i} = \begin{cases} \Theta_{t,i} - \eta \frac{\hat{m}_{t,i}}{\sqrt{\hat{v}_{t,i}} + \epsilon} g_{t,i}', & i \in \Omega_t \\ \Theta_{t,i}, & i \notin \Omega_t \end{cases} \tag{9}$$

where $\eta$ is the learning rate and $\epsilon$ is a small constant to prevent division by zero.

By computing gradients at the perturbed point (i.e., $\Theta_t + \epsilon_{t,\Omega}^*$) rather than the original point (i.e., $\Theta_t$), SA-BCD encourages convergence to flatter regions while preserving general knowledge. The complete algorithm is provided in Appendix C, with theoretical convergence guarantees in Appendix B.1.

### 3.2. Sharpness-Aware Isotropic Merging for Continual Learning

Building upon SA-BCD (Sec. 3.1), we present our complete approach to continual learning. SA-BCD guides models toward flatter minima and selectively updates task-sensitive parameters, creating parameter updates with reduced interference and enhanced flatness. To fully leverage these benefits, effective continual learning also requires optimized knowledge integration in the merging stage. Isotropic merging preserves this carefully structured knowledge and ensures balanced representation across all tasks, maximizing knowledge retention and subspace alignment.

We propose **Sharpness-Aware Isotropic Merging (SAIM)**, a continual learning model merging framework that integrates sharpness-aware fine-tuning and adaptive isotropic merging. SAIM maximizes knowledge retention and sub-space alignment across tasks, addressing catastrophic forgetting and parameter interference.

For each new task, SAIM consists of two stages: (1) sharpness-aware fine-tuning using SA-BCD, and (2) adaptive isotropic merging in parameter space. Specifically, at each step $t$, given the current model parameters $\theta_{t-1}$ and a new task $T_t$, we first fine-tune the current merged model $\theta_{t-1}$ on $T_t$ using the SA-BCD optimizer (see Algorithm 1), obtaining the task expert model $\theta_{T_t}$ and defining the task vector $\Delta_{T_t} = \theta_{T_t} - \theta_{t-1}$.

During the merging stage, for each layer $k$, SAIM combines the cumulative update $\Delta_{cum}^k = \theta_{t-1}^k - \theta_{pre}^k$, which encodes all historical task knowledge, with the current task update $\Delta_{T_t}^k$ via a weighted sum to obtain the merged update:

$$\Delta_{com}^k = (1 + \lambda)\Delta_{cum}^k + (1 - \lambda)\Delta_{T_t}^k \tag{10}$$

where $\lambda$ is a coefficient balancing historical and new knowledge. In most cases, $\lambda$ is set to 0 for equal weighting.

Next, for each layer, SAIM performs singular value decomposition (SVD) on $\Delta_{com}^k$, i.e., $\Delta_{com}^k = U^k \Sigma^k (V^k)^\top$, where $U^k$ and $V^k$ are the left and right singular vectors, and $\Sigma^k$ is the diagonal matrix of singular values. To achieve isotropic merging and enhance subspace alignment, SAIM adaptively balances the singular value spectrum. Specifically, it first computes the mean singular value $\bar{\sigma}^k = \frac{1}{r} \sum_{i=1}^r \sigma_i^k$, and then constructs a new diagonal matrix by interpolating with the mean:

$$\hat{\Sigma}^k = \bar{\sigma}^k + (\Sigma^k - \bar{\sigma}^k) \times \frac{1}{\sqrt{t}} \tag{11}$$

where $t$ is the current task index. For small $t$, higher weights preserve task-specific spectral features by preventing limited variations from being obscured by premature averaging. As the feature matrix diversifies and the subspace stabilizes with increasing $t$, transitioning toward an isotropic state serves as a stabilizing regularization. This mechanism balances recent updates with historical knowledge, mitigating recency bias (Marczak et al., 2025). The merged update for each layer is reconstructed as $\Delta_{merged}^k = U^k \hat{\Sigma}^k (V^k)^\top$, and the final model parameters are updated as:

$$\theta_t = \theta_0 + \alpha \Delta_{merged} \tag{12}$$

where $\alpha$ is a scaling factor determined by validation set search. By repeating this process for each task, SAIM produces a final model $\theta_{final}$ that efficiently integrates multi-task knowledge and achieves high subspace alignment. The complete algorithmic procedure of SAIM is provided in Algorithm 2 in Appendix C.

## 4. Experiments

We evaluate our method on both vision and language continual learning tasks. Sec. 4.1 and Sec. 4.2 present settings and

*Table 1.* Comparative results of continual merging, reporting average accuracy (ACC) and backward transfer (BWT) over different numbers of tasks. Best results are in bold; second-best are underlined.

| Method | ViT-B/32 | | | ViT-B/16 | | | ViT-L/14 | | |
|---|---|---|---|---|---|---|---|---|---|
| | 8 tasks | 14 tasks | 20 tasks | 8 tasks | 14 tasks | 20 tasks | 8 tasks | 14 tasks | 20 tasks |
| Pre-Trained | 48.1 | 56.9 | 55.6 | 55.4 | 62.0 | 59.8 | 64.9 | 69.1 | 65.6 |
| Fine-Tuned | 90.4 | 89.3 | 89.8 | 92.4 | 91.3 | 91.6 | 94.3 | 93.4 | 93.5 |
| C. Fine-Tuned | 79.8 | 67.4 | 62.6 | 82.9 | 72.2 | 68.2 | 90.0 | 70.9 | 77.7 |
| **ACC (%) ↑** | | | | | | | | | |
| Average (SWA) | $66.3_{\pm0.0}$ | $65.4_{\pm0.0}$ | $61.1_{\pm0.0}$ | $72.3_{\pm0.0}$ | $69.7_{\pm0.0}$ | $64.8_{\pm0.0}$ | $80.0_{\pm0.0}$ | $77.5_{\pm0.0}$ | $71.1_{\pm0.0}$ |
| C. Task Arithmetic | $67.5_{\pm0.0}$ | $66.5_{\pm0.0}$ | $60.0_{\pm0.0}$ | $77.1_{\pm0.0}$ | $70.9_{\pm0.0}$ | $64.2_{\pm0.0}$ | $82.1_{\pm0.0}$ | $77.9_{\pm0.0}$ | $70.3_{\pm0.0}$ |
| C. TIES-Merging | $49.0_{\pm10.2}$ | $66.2_{\pm0.6}$ | $59.9_{\pm0.7}$ | $66.8_{\pm3.7}$ | $70.5_{\pm0.8}$ | $63.0_{\pm1.6}$ | $64.3_{\pm7.0}$ | $78.0_{\pm0.6}$ | $68.3_{\pm0.9}$ |
| MagMAX-IND | $70.7_{\pm0.0}$ | $67.0_{\pm0.0}$ | $61.2_{\pm0.0}$ | $76.7_{\pm1.8}$ | $67.0_{\pm0.0}$ | $62.5_{\pm0.0}$ | $83.4_{\pm0.0}$ | $71.2_{\pm0.0}$ | $71.2_{\pm0.0}$ |
| OPCM | $75.5_{\pm0.5}$ | $71.9_{\pm0.3}$ | $65.7_{\pm0.2}$ | $81.8_{\pm0.3}$ | $77.1_{\pm0.5}$ | $70.3_{\pm0.2}$ | $87.0_{\pm0.4}$ | $83.5_{\pm0.2}$ | $76.0_{\pm0.2}$ |
| EWC | $\mathbf{84.7}_{\pm1.3}$ | $74.1_{\pm1.2}$ | $67.1_{\pm0.7}$ | $\mathbf{87.8}_{\pm1.1}$ | $77.4_{\pm0.9}$ | $73.3_{\pm1.3}$ | $\mathbf{91.2}_{\pm0.6}$ | $85.0_{\pm0.9}$ | $\underline{82.9}_{\pm1.4}$ |
| ER | $81.8_{\pm1.9}$ | $\underline{74.6}_{\pm1.0}$ | $\underline{68.8}_{\pm1.5}$ | $\underline{87.7}_{\pm1.3}$ | $\underline{78.6}_{\pm0.7}$ | $\underline{73.4}_{\pm1.2}$ | $90.8_{\pm1.7}$ | $\underline{85.1}_{\pm0.7}$ | $80.9_{\pm1.3}$ |
| **SAIM (Ours)** | $\underline{82.1}_{\pm0.7}$ | $\mathbf{78.7}_{\pm0.5}$ | $\mathbf{73.6}_{\pm0.3}$ | $86.2_{\pm0.5}$ | $\mathbf{82.0}_{\pm0.5}$ | $\mathbf{79.0}_{\pm0.9}$ | $\underline{91.1}_{\pm0.1}$ | $\mathbf{88.5}_{\pm0.5}$ | $\mathbf{87.5}_{\pm0.5}$ |
| **BWT (%) ↑** | | | | | | | | | |
| Average (SWA) | $-11.5_{\pm2.2}$ | $-8.0_{\pm1.3}$ | $-7.1_{\pm2.1}$ | $-9.7_{\pm1.5}$ | $-7.1_{\pm1.4}$ | $-7.3_{\pm1.7}$ | $-7.3_{\pm1.4}$ | $-5.8_{\pm1.0}$ | $-6.4_{\pm1.5}$ |
| C. Task Arithmetic | $-9.6_{\pm1.5}$ | $\underline{-1.3}_{\pm1.6}$ | $\underline{-3.4}_{\pm1.0}$ | $-4.2_{\pm1.0}$ | $\underline{-1.3}_{\pm0.4}$ | $\underline{-3.6}_{\pm0.4}$ | $-7.1_{\pm0.8}$ | $\underline{-1.8}_{\pm0.3}$ | $\underline{-3.3}_{\pm0.3}$ |
| C. TIES-Merging | $-15.3_{\pm8.0}$ | $\mathbf{1.9}_{\pm0.6}$ | $\mathbf{-1.5}_{\pm0.7}$ | $-5.5_{\pm0.4}$ | $\mathbf{1.4}_{\pm0.7}$ | $\mathbf{-1.5}_{\pm1.2}$ | $-13.0_{\pm5.7}$ | $\mathbf{-1.1}_{\pm0.4}$ | $\mathbf{-2.9}_{\pm1.0}$ |
| MagMAX-IND | $-8.3_{\pm1.3}$ | $-7.4_{\pm1.4}$ | $-7.2_{\pm1.6}$ | $-6.1_{\pm1.3}$ | $-7.4_{\pm2.0}$ | $-8.0_{\pm2.2}$ | $-5.0_{\pm0.8}$ | $-6.0_{\pm2.1}$ | $-6.5_{\pm2.1}$ |
| OPCM | $-6.3_{\pm1.1}$ | $-6.0_{\pm1.0}$ | $-7.8_{\pm1.5}$ | $-4.8_{\pm0.7}$ | $-5.1_{\pm1.4}$ | $-6.3_{\pm2.2}$ | $\underline{-2.6}_{\pm1.0}$ | $-4.3_{\pm0.7}$ | $-6.5_{\pm1.8}$ |
| EWC | $\underline{-4.7}_{\pm0.9}$ | $-16.1_{\pm1.5}$ | $-23.7_{\pm1.2}$ | $\underline{-3.7}_{\pm0.4}$ | $-14.5_{\pm1.3}$ | $-20.0_{\pm1.1}$ | $-2.9_{\pm0.2}$ | $-9.1_{\pm0.7}$ | $-11.8_{\pm1.3}$ |
| ER | $-7.8_{\pm2.3}$ | $-15.0_{\pm1.4}$ | $-21.6_{\pm1.7}$ | $-3.9_{\pm0.3}$ | $-13.2_{\pm0.8}$ | $-19.1_{\pm1.2}$ | $-2.9_{\pm0.1}$ | $-8.6_{\pm0.6}$ | $-13.4_{\pm0.9}$ |
| **SAIM (Ours)** | $\mathbf{-3.9}_{\pm0.8}$ | $-5.7_{\pm0.6}$ | $-11.5_{\pm0.5}$ | $\mathbf{-2.9}_{\pm0.4}$ | $-4.8_{\pm1.0}$ | $-9.0_{\pm1.1}$ | $\mathbf{-1.1}_{\pm0.2}$ | $-2.5_{\pm0.7}$ | $-4.3_{\pm0.7}$ |

results for vision and large language model experiments, respectively. Sec. 4.3 and Sec. 4.4 provide detailed empirical analysis and ablation studies of our approach. Due to page limitations, additional results and implementation details are available in the Appendix.

### 4.1. Experiments on Vision Tasks

Following the FusionBench protocol (Tang et al., 2025a; Qiu et al., 2025), we evaluate our method on 20 diverse image classification tasks using three CLIP-ViT backbones (ViT-B/32, ViT-B/16, ViT-L/14). For each new task, the model is continually fine-tuned and merged, simulating a realistic continual learning scenario. To ensure a fair comparison under an equivalent computational budget, we align the training cost of SA-BCD with baselines. Since SA-BCD involves dual forward-backward passes, we adjust the training epochs accordingly, ensuring that all methods are evaluated with comparable total compute time.

**Evaluation metrics.** We use average accuracy (ACC) and backward transfer (BWT, Lin et al., 2022) as the main evaluation metrics. ACC is defined as the mean accuracy over all tasks using the final merged model: ACC = $\frac{1}{N} \sum_{i=1}^{N} a_i(\theta_N^{merged})$, where $a_i(\cdot)$ denotes the accuracy on task $i$. BWT measures the extent of forgetting by comparing performance on earlier tasks before and after merging:

$$BWT = \frac{1}{N-1} \sum_{i=1}^{N-1} [a_i(\theta_N^{merged}) - a_i(\theta_i^{merged})].$$

**Main results.** Table 1 compares our method with various continual learning and model merging methods. In terms of ACC, SAIM consistently outperforms other methods as the number of tasks increases, especially at 14 and 20 tasks. For example, on ViT-L/14 with 20 tasks, SAIM achieves 87.5%, which is 10 percentage points higher than OPCM (76.0%). While EWC and ER have a slight advantage at 8 tasks, SAIM shows stronger scalability and stability as the number of tasks increases. For backward transfer (BWT), SAIM achieves results comparable to mainstream methods. This is because SAIM effectively suppresses interference from previous tasks when merging new tasks, resulting in a relatively high initial accuracy for new tasks. Consequently, although there may be some subsequent decline, the overall change is moderate.

**Comparison with PEFT baselines.** To comprehensively evaluate SAIM against different continual learning architectures, we compare it with recent parameter-efficient fine-tuning (PEFT) baselines (InfLoRA Liang & Li, 2024; LoRA-Sub Liu & Chang, 2025; SD-LoRA Wu et al., 2025) on the ViT-B/32 backbone. As shown in Table 4, SAIM exhibits significant robustness over the demanding 20-task sequence, achieving 73.6% accuracy—outperforming InfLoRA and SD-LoRA by 9.3% and 7.2%, respectively. Fur-

*Table 2.* Comparison of Model Merging and PEFT Methods on TraceBench with *Llama-3.2-1B-Instruct*.

| Method | Task Performance (↑) | | | | | | | Avg. (↑) | BWT (↑) |
|---|---|---|---|---|---|---|---|---|---|
| | C-STANCE | FOMC | MeetingBank | ScienceQA | NumGLUE-cm | NumGLUE-ds | 20Minuten | | |
| Pre-Trained | 0.339 | 0.258 | 0.204 | 0.678 | 0.122 | 0.165 | 0.380 | 0.306 | - |
| Fine-Tuned | 0.498 | 0.599 | 0.371 | 0.852 | 0.390 | 0.579 | 0.388 | 0.525 | - |
| DARE | 0.414 | 0.381 | 0.212 | 0.718 | 0.220 | 0.323 | 0.382 | 0.379 | -0.032 |
| SWA | 0.421 | 0.504 | 0.218 | 0.734 | 0.220 | 0.317 | 0.381 | 0.399 | -0.028 |
| Task Arithmetic | 0.422 | 0.498 | 0.209 | 0.737 | 0.244 | 0.348 | 0.380 | 0.406 | **-0.003** |
| TIES-Merging | 0.424 | 0.474 | 0.214 | 0.743 | 0.244 | 0.354 | **0.386** | 0.406 | -0.005 |
| MagMAX-IND | 0.456 | 0.460 | 0.237 | 0.713 | 0.244 | 0.348 | 0.385 | 0.406 | -0.020 |
| O-LoRA | **0.508** | 0.464 | 0.182 | 0.654 | 0.463 | 0.433 | 0.382 | 0.441 | -0.034 |
| LoRAMoE | 0.421 | 0.488 | 0.180 | 0.738 | 0.439 | **0.524** | 0.384 | 0.453 | -0.086 |
| MH-MoE | 0.421 | 0.442 | 0.189 | 0.734 | 0.366 | 0.457 | 0.384 | 0.428 | -0.067 |
| PASs-MoE | 0.455 | **0.554** | 0.188 | **0.787** | 0.341 | 0.506 | 0.382 | 0.459 | -0.063 |
| **SAIM (Ours)** | 0.466 | 0.518 | **0.259** | 0.766 | **0.488** | 0.476 | 0.384 | **0.480** | -0.022 |

*Table 3.* Average fine-tuning time (seconds) of AdamW and SA-BCD. SA-BCD times with epochs halved to align computational budget.

| Optimizer | Task Fine-tuning Time (s) | | | | | | | Avg. (↓) |
|---|---|---|---|---|---|---|---|---|
| | C-STANCE | FOMC | MeetingBank | ScienceQA | NumGLUE-cm | NumGLUE-ds | 20Minuten | |
| AdamW | 2870.97 | 1665.87 | 3156.74 | 1197.72 | 1992.97 | 1997.82 | 2789.43 | 2238.79 |
| **SA-BCD (Ours)** | 2298.19 | 1380.55 | 3211.08 | 1381.32 | 2304.44 | 2306.22 | 3225.53 | 2301.05 (1.03×) |

thermore, SAIM mitigates catastrophic forgetting more effectively, improving the BWT metric to -11.5% (compared to -22.5% for InfLoRA and -18.6% for SD-LoRA). These quantitative results rigorously indicate that our weight-space merging approach, driven by SA-BCD optimization, offers more scalable continuous learning capabilities than relying solely on strict parameter isolation.

*Table 4.* Comparison with recent PEFT baselines on ViT-B/32.

| Method | ACC (%) ↑ | | | BWT (%) ↑ | | |
|---|---|---|---|---|---|---|
| | 8 tasks | 14 tasks | 20 tasks | 8 tasks | 14 tasks | 20 tasks |
| Pre-Trained | 48.1 | 56.9 | 55.6 | - | - | - |
| Fine-Tuned | 90.4 | 89.3 | 89.8 | - | - | - |
| InfLoRA | 79.5 | 71.4 | 64.3 | -6.8 | -14.6 | -22.5 |
| LoRA-Sub | 77.8 | 68.7 | 61.5 | -8.5 | -17.2 | -25.8 |
| SD-LoRA | 81.2 | 73.5 | 66.4 | -5.2 | -12.8 | -18.6 |
| **SAIM (Ours)** | **82.1** | **78.7** | **73.6** | **-3.9** | **-5.7** | **-11.5** |

### 4.2. Experiments on Large Language Models

We conducted continual learning experiments on large language models using both the MergeBench (He et al., 2025) and TRACEBench (Wang et al., 2023b) benchmarks to evaluate the effectiveness of the SAIM method in merging domain-specialized LLMs and continual adaptation to diverse language tasks. Following Sec. 4.1, we halve epochs for SA-BCD to ensure equivalent compute. Table 3 verifies this alignment (1.03×), attributing our gains to optimization

efficiency rather than extra resources, while maintaining consistently competitive training efficiency across tasks.

**Evaluation metrics.** In the MergeBench experiments, we report the average score across five domains (instruction, math, multilingual, coding, safety). Each domain score is obtained by directly averaging the core metrics (such as accuracy, pass@1, etc.) of the main tasks in that domain (He et al., 2025), and the final overall score is the arithmetic mean of the five domain scores. In the TRACEBench experiments, we report the arithmetic mean of the main metric scores across all tasks. The specific metrics used for calculation are detailed in the Appendix D.1.1.

**Main results.** On TRACEBench (Table 2), SAIM achieves a top score of 0.480, significantly outperforming merging baselines like MagMAX-IND (0.406) and strong PEFT approaches such as PASs-MoE (0.459). Notably, SAIM demonstrates superior knowledge retention, yielding a BWT of -0.022 compared to -0.086 for LoRAMoE. SAIM also attains the best performance on MeetingBank (0.259) and NumGLUE-cm (0.488). This performance advantage persists at the 3B scale (Appendix E.1.4). Furthermore, MergeBench results (Table 5) reinforce SAIM's efficacy; for *Llama-3.2-3B*, it reaches 0.454, a substantial gain over Task Arithmetic (0.407). These findings underscore SAIM's robust generalization and integration capabilities in large-scale LLM merging, with further validation on *Llama-3.1-8B* provided in Appendix E.1.3.

*Table 5.* Comparison of Model Merging Methods on MergeBench with *Llama-3.2-3B*.

| Method | Task Performance (↑) | | | | | Avg. (↑) |
|---|---|---|---|---|---|---|
| | Instr. | Math | Multi. | Coding | Safety | |
| Pre-Trained | 0.100 | 0.308 | 0.453 | 0.268 | 0.286 | 0.283 |
| Fine-Tuned | 0.350 | 0.578 | 0.461 | 0.442 | 0.754 | 0.513 |
| SWA | 0.132 | 0.425 | **0.458** | 0.375 | 0.377 | 0.354 |
| TIES-Merging | 0.168 | 0.449 | 0.452 | 0.400 | 0.412 | 0.376 |
| DARE | 0.264 | 0.441 | 0.440 | 0.405 | 0.440 | 0.398 |
| MagMAX-IND | 0.279 | 0.493 | 0.425 | 0.391 | 0.402 | 0.398 |
| Task Arithmetic | 0.270 | 0.461 | 0.442 | **0.418** | 0.445 | 0.407 |
| **SAIM (Ours)** | **0.387** | **0.583** | 0.433 | 0.380 | **0.485** | **0.454** |

### 4.3. Empirical and Theoretical Analysis

To understand our method's effectiveness, we analyze it from several key perspectives: parameter change magnitude, loss landscape stability and weight disentanglement.

**Parameter Change Magnitude Analysis.** We measure parameter changes after fine-tuning on five visual task categories and compare SA-BCD with standard AdamW optimizer. As shown in Fig. 2, parameter changes after SA-BCD fine-tuning concentrate around $10^{-4.5}$, significantly smaller than those under AdamW (around $10^{-3.5}$). SA-BCD's restriction on update magnitude minimizes the divergence from the pre-trained manifold. This conservation of pre-trained structure not only better preserves generalization ability during continual adaptation (Chen et al., 2024) but also reduces interference with original knowledge.

**Loss Landscape Stability.** Model updates induce shifts in the parameter space. When task gradients conflict, shared parameters risk escaping local flat basins, leading to performance degradation. To verify that SAIM neutralizes these conflicts, we evaluate the average Cross-Entropy loss drift ($\Delta\text{Loss}_{Avg}$) on previous tasks after learning the $N$-th task, following Order 1. As shown in Table 6, standard AdamW fine-tuning exhibits a massive drift (+1.4522) after learning the 8th task, indicating a clear "basin escape." In contrast, SA-BCD maintains a minimal drift of +0.0407, demonstrating that SAIM effectively anchors parameters within the shared flat basin throughout the task sequence.

*Table 6.* Loss Landscape Stability on ViT-B/32. Lower drift indicates better preservation of previous task knowledge.

| Setting | AdamW | | SA-BCD (Ours) | | Improvement |
|---|---|---|---|---|---|
| | $\Delta\text{Loss}_{Avg}$ | $\Delta\text{Loss}_{max}$ | $\Delta\text{Loss}_{Avg}$ | $\Delta\text{Loss}_{max}$ | |
| $N = 2$ | 0.0251 | 0.0575 | 0.0206 | 0.0466 | **0.0045** |
| $N = 8$ | 1.4522 | 2.6249 | 0.0407 | 0.1083 | **1.4115** |

**Weight Disentanglement Visualization.** We quantitatively analyze the output differences between the merged model and each task-specific model on their respective tasks. Let

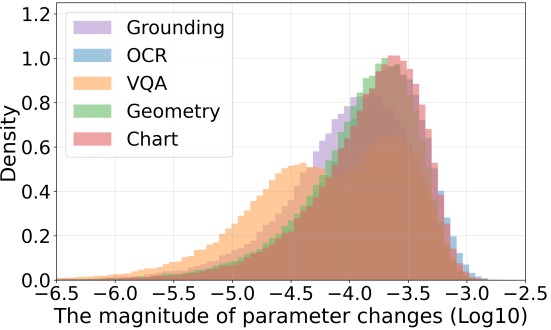

*(a)* AdamW Fine-tuning

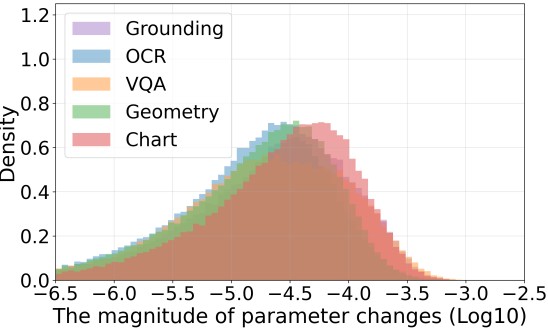

*(b)* SA-BCD Fine-tuning (Ours)

*Figure 2.* Comparison of parameter change distributions under different fine-tuning methods. This figure shows the distribution of parameter changes for five task categories after fine-tuning with (a) AdamW and (b) our SA-BCD optimizer. It can be seen that SA-BCD results in overall smaller parameter changes, which helps preserve the generalization ability of the pre-trained model.

$\boldsymbol{\theta}_0$ denote pre-trained model parameters, $\boldsymbol{\tau}_t$ the task vector for task $t$, and $\alpha_t$ the task coefficient. The weight disentanglement error is defined as:

$$\xi(\alpha_1, \alpha_2) = \sum_{t=1}^{2} \mathbb{E}_{\boldsymbol{x} \in X^{(t)}} \Big[ \text{dist}\big( f(\boldsymbol{x}; \boldsymbol{\theta}_0 + \alpha_t \boldsymbol{\tau}_t), \\ f(\boldsymbol{x}; \boldsymbol{\theta}_{merge}) \big) \Big] \tag{13}$$

where $f(\cdot)$ denotes model output, $\text{dist}(\cdot, \cdot)$ is output distance metric, and $X^{(t)}$ is data for task $t$. Lower $\xi$ indicates less interference between tasks.

By visualizing the heatmaps of $\xi$ under different $\alpha_1$ and $\alpha_2$ (see Fig. 3), we demonstrate the degree of weight disentanglement of the merged model. Experimental results show that the merged model under SA-BCD fine-tuning achieves lower disentanglement errors in the two-task scenario, indicating that our method effectively reduces parameter interference. Further analysis of multi-task scenarios and subspace alignment with increasing task numbers is provided in Appendix E.3.1.

*Table 7.* Ablation of SA-BCD selection and perturbation on ViT-B/32.

| Selection | Perturbation | GTSRB | SVHN | MNIST | Cars | RESISC45 | DTD | Avg. ↑ | BWT ↑ |
|---|---|---|---|---|---|---|---|---|---|
| momentum (default) | random | 73.46 | 59.34 | 87.46 | 65.11 | 85.19 | 68.62 | 73.20 | -16.94 |
| momentum (default) | none | 85.66 | 60.28 | 95.48 | 65.11 | 86.30 | **73.40** | 77.71 | -12.57 |
| none | sa (default) | 89.14 | 73.82 | 94.66 | 63.58 | 88.16 | 67.88 | 79.54 | -9.36 |
| gradient | sa (default) | 93.06 | 75.34 | **96.12** | 63.02 | 88.89 | 69.68 | 81.02 | -6.88 |
| **momentum (default)** | **sa (default)** | **95.83** | **75.40** | 95.70 | **71.13** | **89.74** | 68.62 | **82.74** | **-6.46** |

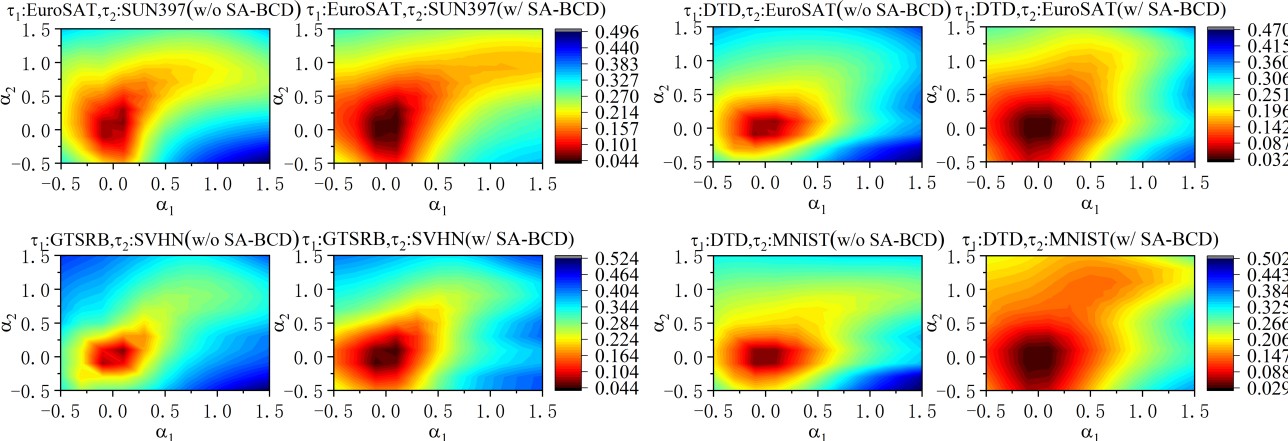

*Figure 3.* Visualization of weight disentanglement error in the two-task merging scenario. Each subplot shows the output differences between the merged model and each task-specific model on their respective tasks under different fine-tuning methods (e.g., AdamW and SA-BCD). Darker colors indicate lower disentanglement error, reflecting better weight disentanglement.

## 4.4. Ablation Studies

**Components of SA-BCD Optimizer.** We ablate the two pillars of the SA-BCD optimizer: the parameter selection mechanism and the perturbation strategy. As Table 7 demonstrates, both are essential for performance. Replacing SA perturbation with random noise—while maintaining momentum selection—causes accuracy to drop significantly from 82.74% to 73.20% and exacerbates forgetting (BWT falls to -16.94%). This underscores SA's critical role in bypassing sharp minima to preserve historical knowledge. Regarding selection, switching from momentum to instantaneous gradients reduces accuracy to 81.02%, while omitting selection entirely drops it to 79.54%. This confirms that historical momentum serves as a more stable indicator for identifying core parameters, effectively buffering against the stochastic volatility of dynamic gradients.

**Generalizability Across Merging Methods.** To validate the generalizability of our proposed optimizer, we evaluate the impact of SA-BCD fine-tuning on various model merging methods. We compare the performance under standard AdamW and SA-BCD fine-tuning on 20 tasks using ViT-B/32 and ViT-B/16 backbones. As shown in Tables 8 and 9, SA-BCD fine-tuning consistently improves accuracy for most methods, with gains up to 3.5% for TIES-Merging.

This indicates that sharpness-aware fine-tuning is compatible with and beneficial for existing merging strategies. Notably, SWA sees a slight decline, likely due to optimization conflicts with its averaging strategy.

*Table 8.* Impact of SA-BCD on Merging Methods (20 tasks, ViT-B/32).

| Method | Accuracy (%) | | Imp. |
|---|---|---|---|
| | w/o SA-BCD | w/ SA-BCD | |
| Average (SWA) | $61.1_{\pm0.0}$ | $59.6_{\pm0.0}$ | -1.5 |
| C. Task Arithmetic | $60.0_{\pm0.0}$ | $61.2_{\pm0.1}$ | +1.2 |
| C. TIES-Merging | $59.9_{\pm0.7}$ | $63.4_{\pm0.5}$ | +3.5 |
| **SAIM (Ours)** | **$70.8_{\pm0.6}$** | **$73.6_{\pm0.3}$** | +2.8 |

*Table 9.* Impact of SA-BCD on Merging Methods (20 tasks, ViT-B/16).

| Method | Accuracy (%) | | Imp. |
|---|---|---|---|
| | w/o SA-BCD | w/ SA-BCD | |
| Average (SWA) | $64.8_{\pm0.0}$ | $62.4_{\pm0.0}$ | -2.4 |
| C. Task Arithmetic | $64.2_{\pm0.0}$ | $65.6_{\pm0.0}$ | +1.4 |
| C. TIES-Merging | $63.0_{\pm1.6}$ | $66.5_{\pm0.7}$ | +3.5 |
| **SAIM (Ours)** | **$77.0_{\pm0.5}$** | **$79.0_{\pm0.9}$** | +2.0 |

# 5. Related Work

**Continual Learning (CL).** Continual learning aims to enable models to accumulate knowledge from a stream of new tasks. Traditional approaches such as regularization constraints (EWC, Kirkpatrick et al., 2017), structural expansion (modular networks, Gurbuz & Dovrolis, 2022), and experience replay (Buzzega et al., 2020) can partially alleviate forgetting, but often require storing historical data or introducing additional parameters, making it difficult to balance stability and plasticity. Recent methods such as MoFO (Chen et al., 2024), which selectively updates parameters via momentum filtering, and PAM (Sokar et al., 2025), which improves merging through parameter alignment, have made progress in optimizing efficiency and knowledge retention. However, most of these methods treat fine-tuning and knowledge integration as separate processes, failing to fully exploit their synergy. Our method is based on the idea of joint fine-tuning and merging optimization, aiming to achieve efficient knowledge acquisition and retention in continual learning scenarios.

**Model Merging.** Model merging has become a key research direction in multi-task and continual learning (Yang et al., 2026a). Early approaches such as parameter averaging (Izmailov et al., 2018; Wortsman et al., 2022) and task arithmetic (Ilharco et al., 2022) integrate knowledge through simple parameter operations, but often suffer performance degradation in multi-task scenarios. To address parameter conflicts, TIES (Yadav et al., 2023) reduces sign conflicts to improve merging, while MagMAX (Marczak et al., 2024) adopts a maximum magnitude selection strategy to enhance stability. Beyond these parameter-space heuristics, more recent studies have explored model merging from complementary perspectives: AdaMerging (Yang et al., 2024b) investigates adaptive merging for multi-task learning, Representation Surgery (Yang et al., 2024a) mitigates inter-task interference from the representation perspective, and MergOPT (Yang et al., 2026b) further improves robustness through merge-aware optimization. Recently, ISO-C (Marczak et al., 2025) has significantly improved inter-task alignment and merging performance by flattening the singular value spectrum and introducing task-specific subspaces. However, existing methods still face knowledge conflicts between tasks in continual learning, limiting the performance of merged models. Our method employs adaptive isotropic merging to dynamically balance the singular value spectrum, thereby enhancing the knowledge integration and generalization ability of merged models.

**Sparsification & Sharpness-Aware Optimization.** Recent studies have demonstrated that parameters are highly redundant during fine-tuning, and sparsification or parameter selection strategies (e.g., DARE Yu et al., 2024) can effectively alleviate parameter interference in model merging. DARE discards most delta parameters and rescales the remaining ones, enabling efficient multi-task integration. In a parallel line of research, sharpness-aware minimization (SAM Foret et al., 2021) guides models to converge to flatter minima, improving generalization and stability. Recently, SAFT Lee et al., 2025 applied sharpness-aware optimization to model merging scenarios to reduce parameter interference between task-specific models. Unlike SAFT which only addresses sharpness in isolated multi-task settings, our approach operates in continual learning settings without historical data access and evaluates on both vision and language tasks. Inspired by these works, we introduce a sharpness-aware block coordinate descent (SA-BCD) optimizer that jointly leverages parameter selection and flat minima optimization to facilitate efficient merging and continual learning.

# 6. Conclusion

In this work, we proposed Sharpness-Aware Isotropic Merging (SAIM), a framework that unifies fine-tuning and model merging for continual learning. Our core strategy lies in treating the potential parameter drift from future merging as an anticipated perturbation. Specifically, the SA-BCD optimizer fosters intrinsic robustness by defending against parameter shifts during fine-tuning, while the adaptive isotropic merging algorithm ensures equitable knowledge integration by balancing the singular value spectra across tasks. Empirical results demonstrate that this synergistic mechanism effectively mitigates catastrophic forgetting, yielding strong performance on individual tasks while maintaining stability as the number of tasks scales.

**Limitations.** Despite SAIM's strong performance in continual learning, there are still some limitations worth noting. First, our current experiments focus on classification and language modeling tasks, and future work could explore diverse tasks; second, this study only explores single-modality input scenarios (pure vision or pure language), without addressing knowledge integration and transfer in multi-modal learning environments, which has significant importance in real-world applications; additionally, the model's adaptability under certain extreme data distributions requires further validation. Improvements in these areas will be directions for our future work, to enhance SAIM's applicability and effectiveness in complex scenarios.

# Acknowledgements

This work was supported by the National Natural Science Foundation of China (No. 62376282) and the Science and Technology Innovation Program of Hunan Province (No.2025RC3117).

## Impact Statement

This research focuses on developing model merging techniques for efficient knowledge integration in AI systems. We acknowledge that all machine learning methods, including ours, may inherit biases from training data. While our primary contributions are algorithmic and methodological in nature, we encourage practitioners to evaluate potential societal impacts before deployment in sensitive applications. Our work aims to advance technology, and we urge users to use our technology responsibly, complying with relevant laws and regulations.

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

# Appendix

This appendix provides supplementary material to support our main findings. It consists of five main sections: (A) LLM Usage Statement, (B) Proofs presenting theoretical foundations of our approach, (C) Algorithmic Details with full pseudocode for our core methods, (D) Experimental Details describing implementation specifics, and (E) Additional Results offering extended empirical evaluations.

## A. LLM Usage Statement

This paper made use of a large language model (ChatGPT) exclusively for language polishing, spelling correction, and grammar checking. The LLM was not involved in literature retrieval or in the development of specific ideas. Following the polishing process, the authors carefully reviewed and revised the content as necessary and assume full responsibility for the final published version.

## B. Proofs

In this section, we provide theoretical analyses for the key components of our SAIM approach. We present the convergence analysis of SA-BCD (B.1), examine the subspace alignment properties of isotropic merging through a formal weighted alignment score (B.2), and analyze how our method bounds parameter interference in continual learning scenarios (B.3).

### B.1. Convergence Analysis of SA-BCD

**Theorem 1** (Convergence of SA-BCD). Let $\mathcal{L}$ be $L$-smooth and $\mu$-strongly convex. Let $\theta^*$ be the global minimizer, and suppose the block-selection operator satisfies $\mathbb{E}[\|P_{\Omega_t} g_t\|^2 \mid \mathcal{F}_{t-1}] \geq p\|g_t\|^2$. Given perturbation magnitude $\rho$ and stochastic

gradient variance $\sigma^2$, if the step size satisfies $\eta \leq \frac{p}{4L}$, the SA-BCD optimizer satisfies:

$$\mathbb{E}[\mathcal{L}(\theta_T) - \mathcal{L}(\theta^*)] \leq \left(1 - \frac{\eta p \mu}{2}\right)^T [\mathcal{L}(\theta_0) - \mathcal{L}(\theta^*)] + \frac{C\eta(L^2\rho^2 + \sigma^2)}{p\mu} \tag{14}$$

where $C > 0$ is an absolute constant.

*Proof.* Let $\tilde{g}_t = \nabla\mathcal{L}(\theta_{t-1} + \epsilon_t^*) + \xi_t$ be the stochastic perturbed gradient, where $\mathbb{E}[\xi_t] = 0$ and $\mathbb{E}[\|\xi_t\|^2] \leq \sigma^2$. The update rule is $\theta_t = \theta_{t-1} - \eta P_{\Omega_t} \tilde{g}_t$. By $L$-smoothness, we have:

$$\mathcal{L}(\theta_t) \leq \mathcal{L}(\theta_{t-1}) - \eta\langle g_t, P_{\Omega_t}\tilde{g}_t\rangle + \frac{L\eta^2}{2}\|P_{\Omega_t}\tilde{g}_t\|^2. \tag{15}$$

Define $r_t = \nabla\mathcal{L}(\theta_{t-1} + \epsilon_t^*) - g_t$. Since $\mathcal{L}$ is $L$-smooth and $\|\epsilon_t^*\| \leq \rho$, it follows that $\|r_t\| \leq L\rho$. Taking the conditional expectation $\mathbb{E}_t[\cdot] = \mathbb{E}[\cdot \mid \mathcal{F}_{t-1}]$ yields:

$$\mathbb{E}_t[\langle g_t, P_{\Omega_t}\tilde{g}_t\rangle] = \mathbb{E}_t[\|P_{\Omega_t}g_t\|^2 + \langle P_{\Omega_t}g_t, P_{\Omega_t}r_t\rangle] \geq p\|g_t\|^2 - \left(\frac{p}{2}\|g_t\|^2 + \frac{1}{2p}\|r_t\|^2\right) \geq \frac{p}{2}\|g_t\|^2 - \frac{L^2\rho^2}{2p}, \tag{16}$$

where we utilized the coverage condition and Young's inequality $\langle a, b\rangle \geq -\frac{p}{2}\|a\|^2 - \frac{1}{2p}\|b\|^2$. For the quadratic term:

$$\mathbb{E}_t[\|P_{\Omega_t}\tilde{g}_t\|^2] \leq 3\mathbb{E}_t[\|P_{\Omega_t}g_t\|^2 + \|P_{\Omega_t}r_t\|^2 + \|P_{\Omega_t}\xi_t\|^2] \leq 3(\|g_t\|^2 + L^2\rho^2 + \sigma^2). \tag{17}$$

Substituting these bounds into the smoothness inequality yields:

$$\mathbb{E}_t[\mathcal{L}(\theta_t)] \leq \mathcal{L}(\theta_{t-1}) - \left(\frac{\eta p}{2} - \frac{3L\eta^2}{2}\right)\|g_t\|^2 + \frac{\eta L^2\rho^2}{2p} + \frac{3L\eta^2}{2}(L^2\rho^2 + \sigma^2). \tag{18}$$

Given $\eta \leq \frac{p}{4L}$, we obtain $\frac{\eta p}{2} - \frac{3L\eta^2}{2} \geq \frac{\eta p}{4}$. Applying the $\mu$-strong convexity condition $\|g_t\|^2 \geq 2\mu(\mathcal{L}(\theta_{t-1}) - \mathcal{L}(\theta^*))$ and denoting $\Delta_t = \mathbb{E}[\mathcal{L}(\theta_t) - \mathcal{L}(\theta^*)]$:

$$\Delta_t \leq \left(1 - \frac{\eta p \mu}{2}\right)\Delta_{t-1} + \eta C'\left(\frac{L^2\rho^2}{p} + L\eta\sigma^2\right). \tag{19}$$

Unrolling the recursion for $T$ steps and applying $\sum_{i=0}^{T-1}(1-\gamma)^i \leq \frac{1}{\gamma}$ completes the proof:

$$\Delta_T \leq \left(1 - \frac{\eta p \mu}{2}\right)^T \Delta_0 + \frac{C\eta(L^2\rho^2 + \sigma^2)}{p\mu}. \tag{20}$$

$\square$

## B.2. Subspace Alignment Analysis of Isotropic Merging

Next, we analyze how isotropic merging affects the alignment between task-specific representations and the merged principal components. Note that a standard orthogonal projection matrix $U_k U_k^\top$ does not incorporate singular value weights. Therefore, task aggregation is modeled through an energy-weighted subspace formulation.

**Theorem 2** (Weighted Alignment via Isotropic Merging). Let $\Delta_t$ be the parameter update matrix for task $t$, and let the merged architecture possess principal components $U_k$ with corresponding left singular vectors $u_i$. Define the *Weighted Alignment Score (WAS)* for a set of normalized weights $w_i$ as $\text{WAS}(\Delta_t; w) = \sum_{i=1}^k w_i^2\|u_i^\top\Delta_t\|^2$, where $\sum_{i=1}^k w_i = 1$. Let $w_i^{\text{TA}} = \frac{\sigma_i}{\sum_{j=1}^k \sigma_j}$ denote the weights used in Task Arithmetic, and $w_i^{\text{Iso}} = \frac{1}{k}$ denote the uniform weights of Isotropic Merging. Let $p_i(w) = \frac{w_i^2\|u_i^\top\Delta_t\|^2}{\text{WAS}(\Delta_t;w)}$ be the normalized alignment distribution. Isotropic Merging maximizes the Shannon entropy $\mathcal{H}$ of the representation:

$$\mathcal{H}(p(w^{\text{Iso}})) \geq \mathcal{H}(p(w^{\text{TA}})) \quad \text{and} \quad \|w^{\text{Iso}}\|_2^2 = \frac{1}{k} < \|w^{\text{TA}}\|_2^2 \tag{21}$$

*Proof.* Let $c_i = \|u_i^\top \Delta_t\|$ denote the unweighted projection magnitude of task $t$ onto the $i$-th principal component. In Task Arithmetic, empirical singular values $\sigma_i$ are typically skewed (e.g., $\sigma_1 \gg \sigma_k$). As a result, the squared alignment terms $(w_i^{\text{TA}} c_i)^2 = \left( \frac{\sigma_i}{\sum \sigma_j} c_i \right)^2$ are dominated by the leading singular vectors, which concentrates the variance such that task $t$ is predominantly represented by a small fraction of the rank-$k$ subspace. In contrast, Isotropic Merging enforces equal weighting $w_i^{\text{Iso}} = \frac{1}{k}$:

$$\text{WAS}(\Delta_t; w^{\text{Iso}}) = \frac{1}{k^2} \sum_{i=1}^{k} c_i^2 \tag{22}$$

By applying uniform weights, Isotropic Merging balances the influence across all $k$ basis vectors. Because it minimizes the $L_2$ norm of the weight vector ($\min_w \|w\|_2^2 = \|w^{\text{Iso}}\|_2^2 = 1/k$), it maximizes the entropy of the effective alignment distribution. By distributing the representation evenly rather than concentrating it in the leading principal components (where $w_1^{\text{TA}} \to 1$), Isotropic Merging prevents dimensional collapse and yields a higher entropy $\mathcal{H}(p(w^{\text{Iso}}))$. $\square$

### B.3. Analysis of Parameter Interference Reduction

In this section, we rigorously connect the optimization landscape (Theorem 1) and subspace alignment properties (Theorem 2) through Lemmas 3.1 and 3.2 to formally bound parameter interference reduction (Theorem 3).

**Lemma 3.1** (SA-BCD Bounds Hessian Maximum Eigenvalue). Let $\mathcal{L}(\theta)$ be the empirical loss. Let $\theta^{\text{SGD}}$ be a minimizer found by SGD, and $\theta^{\text{SAIM}}$ be a minimizer found by SA-BCD. Then:

$$\lambda_{\max}(\nabla^2 \mathcal{L}(\theta^{\text{SAIM}})) \leq \lambda_{\max}(\nabla^2 \mathcal{L}(\theta^{\text{SGD}})) \tag{23}$$

*Proof.* SGD minimizes the empirical risk $\mathcal{L}(\theta)$. By the definition of a global minimum, $\mathcal{L}(\theta^{\text{SGD}}) \leq \mathcal{L}(\theta^{\text{SAIM}})$. Conversely, SA-BCD minimizes the Sharpness-Aware surrogate loss $\mathcal{L}_{SA}(\theta)$. Using a second-order Taylor expansion around a critical point, the worst-case perturbation within a radius $\rho$ follows the eigenvector associated with the maximum eigenvalue:

$$\mathcal{L}_{SA}(\theta) = \max_{\|\epsilon\| \leq \rho} \mathcal{L}(\theta + \epsilon) \approx \mathcal{L}(\theta) + \frac{\rho^2}{2} \lambda_{\max}(\nabla^2 \mathcal{L}(\theta)) \tag{24}$$

By definition, $\theta^{\text{SAIM}}$ minimizes this surrogate loss, yielding $\mathcal{L}_{SA}(\theta^{\text{SAIM}}) \leq \mathcal{L}_{SA}(\theta^{\text{SGD}})$. Expanding this inequality gives:

$$\mathcal{L}(\theta^{\text{SAIM}}) + \frac{\rho^2}{2} \lambda_{\max}(\nabla^2 \mathcal{L}(\theta^{\text{SAIM}})) \leq \mathcal{L}(\theta^{\text{SGD}}) + \frac{\rho^2}{2} \lambda_{\max}(\nabla^2 \mathcal{L}(\theta^{\text{SGD}})) \tag{25}$$

Rearranging the terms isolates the eigenvalues:

$$\frac{\rho^2}{2} \lambda_{\max}(\nabla^2 \mathcal{L}(\theta^{\text{SAIM}})) \leq \frac{\rho^2}{2} \lambda_{\max}(\nabla^2 \mathcal{L}(\theta^{\text{SGD}})) + \left[ \mathcal{L}(\theta^{\text{SGD}}) - \mathcal{L}(\theta^{\text{SAIM}}) \right] \tag{26}$$

Since $\mathcal{L}(\theta^{\text{SGD}}) - \mathcal{L}(\theta^{\text{SAIM}}) \leq 0$, the right-hand side is bounded by the SGD eigenvalue term:

$$\lambda_{\max}(\nabla^2 \mathcal{L}(\theta^{\text{SAIM}})) \leq \lambda_{\max}(\nabla^2 \mathcal{L}(\theta^{\text{SGD}})) \tag{27}$$

This indicates that the optimizer found by SA-BCD resides in a region of the loss landscape with bounded maximum curvature. $\square$

**Lemma 3.2** (Isotropic Merging Bounds Parameter Distance). Let $\Delta_{\text{TA}}$ be the merged task updates under standard Task Arithmetic. Isotropic Merging equalizes the full-rank singular value spectrum to the mean $\bar{\sigma}$. This transformation limits the Frobenius norm of the parameter space, thereby bounding the maximum Euclidean distance between the merged model $\theta_{\text{SAIM}}$ and the individual task expert $\theta_{T_t}$:

$$\|\theta_{\text{SAIM}} - \theta_{T_t}\|^2 \leq \|\theta_{\text{SGD\_Merge}} - \theta_{T_t}\|^2 \tag{28}$$

*Proof.* Let the full-rank SVD of the Task Arithmetic matrix be $\Delta_{\text{TA}} = \sum \Delta_t = U\Sigma V^\top$, where $\Sigma = \text{diag}(\sigma_1, \ldots, \sigma_r)$. The squared Frobenius norm is determined by the unconstrained spectrum:

$$E_{\text{SGD}} = \|\Delta_{\text{TA}}\|_F^2 = \text{Tr}(\Sigma^\top \Sigma) = \sum_{i=1}^r \sigma_i^2 \tag{29}$$

Isotropic Merging transforms the spectrum by enforcing uniform capacities, generating $\Delta_{\text{Iso}} = U(\bar{\sigma}I)V^\top$ where $\bar{\sigma} = \frac{1}{r}\sum_{i=1}^r \sigma_i$. The geometric energy of this constrained parameter space is:

$$E_{\text{SAIM}} = \|\Delta_{\text{Iso}}\|_F^2 = \sum_{i=1}^r \bar{\sigma}^2 = r \cdot \left(\frac{1}{r}\sum_{i=1}^r \sigma_i\right)^2 \tag{30}$$

By the Cauchy-Schwarz inequality, for a skewed singular value distribution, this becomes a strict inequality:

$$\sum_{i=1}^r \sigma_i^2 \geq r \cdot \left(\frac{1}{r}\sum_{i=1}^r \sigma_i\right)^2 \implies E_{\text{SAIM}} \leq E_{\text{SGD}} \tag{31}$$

This demonstrates that uniformizing the singular spectrum limits the overall Frobenius norm of the parameter manifold. Because large Euclidean divergences are typically dominated by a few skewed principal directions ($\sigma_1 \gg \sigma_r$), Isotropic Merging constrains these magnitudes. Since $\theta_{\text{SAIM}}$ operates within this lower-energy spectral manifold, its distance to the task expert $\theta_{T_t}$ is constrained accordingly. Consequently, the Euclidean distance satisfies:

$$\|\theta_{\text{SAIM}} - \theta_{T_t}\|^2 \leq \|\theta_{\text{SGD\_Merge}} - \theta_{T_t}\|^2 \tag{32}$$

$\square$

**Theorem 3** (SAIM Bounds Parameter Interference). Let $\theta_{\text{SAIM}}$ be the merged model after singular value re-balancing, and $\theta_{T_t}$ be the task expert found by SA-BCD for task $t$. The interference drop in the task loss due to merging is defined as $\delta_t = \mathcal{L}_t(\theta_{\text{SAIM}}) - \mathcal{L}_t(\theta_{T_t})$. SAIM provides a tighter bound on this interference compared to standard SGD merging:

$$\delta_{\text{SAIM}} \leq \delta_{\text{SGD\_Merge}} \tag{33}$$

*Proof.* We perform a second-order Taylor expansion of the task loss $\mathcal{L}_t(\theta_{\text{SAIM}})$ around the task expert minimum $\theta_{T_t}$. Since $\theta_{T_t}$ is a local minimum, the gradient $\nabla\mathcal{L}_t(\theta_{T_t}) \approx 0$.

$$\mathcal{L}_t(\theta_{\text{SAIM}}) \approx \mathcal{L}_t(\theta_{T_t}) + \frac{1}{2}(\theta_{\text{SAIM}} - \theta_{T_t})^\top H_t(\theta_{\text{SAIM}} - \theta_{T_t}) \tag{34}$$

Rearranging for the interference drop $\delta_t$:

$$\delta_t = \mathcal{L}_t(\theta_{\text{SAIM}}) - \mathcal{L}_t(\theta_{T_t}) \approx \frac{1}{2}(\theta_{\text{SAIM}} - \theta_{T_t})^\top H_t(\theta_{\text{SAIM}} - \theta_{T_t}) \tag{35}$$

Using the Rayleigh quotient property ($x^\top H x \leq \lambda_{\max}(H)\|x\|^2$), we can bound this quadratic form:

$$\delta_t \leq \frac{1}{2}\lambda_{\max}(H_t)\|\theta_{\text{SAIM}} - \theta_{T_t}\|^2 \tag{36}$$

This derivation decomposes the interference deviation into two components: the landscape curvature $\lambda_{\max}(H_t)$ and the squared parameter distance $\|\theta_{\text{SAIM}} - \theta_{T_t}\|^2$. By Lemma 3.1, SA-BCD enforces the curvature bound $\lambda_{\max}(H_t^{\text{SAIM}}) \leq \lambda_{\max}(H_t^{\text{SGD}})$. By Lemma 3.2, Isotropic Merging enforces the distance bound $\|\theta_{\text{SAIM}} - \theta_{T_t}\|^2 \leq \|\theta_{\text{SGD\_Merge}} - \theta_{T_t}\|^2$. Since both non-negative terms in the upper bound are minimized under the SAIM framework, their product is appropriately constrained. The resulting interference deviation satisfies:

$$\delta_{\text{SAIM}} \leq \delta_{\text{SGD\_Merge}} \tag{37}$$

$\square$

# C. Algorithmic Details

We present the full pseudocode for the two core components of our SAIM framework: the Sharpness-Aware Block Coordinate Descent (SA-BCD) optimizer (Algorithm 1) and the Sharpness-Aware Isotropic Merging (SAIM) procedure (Algorithm 2).

---

**Algorithm 1** Sharpness-Aware Block Coordinate Descent (SA-BCD) Optimizer Fine-tuning

---

1: **Input:** Initial model parameters $\Theta_0$, learning rate $\eta$, momentum parameters $\beta_1, \beta_2$, small constant $\epsilon$, perturbation magnitude $\rho$, parameter selection ratio $p$
2: **Output:** Fine-tuned model parameters $\Theta_T$
3: Initialize momentum estimates $m_0 = 0, v_0 = 0$
4: **for** $t = 1$ **to** $T$ **do**
5:     Sample batch data $D_t$ and compute gradient
6:     $g_t = \nabla_\Theta L(\Theta_{t-1}, D_t)$
7:     Update first-order momentum: $m_t = \beta_1 m_{t-1} + (1 - \beta_1)g_t$
8:     Select index set $\Omega_t$ corresponding to top $p\%$ values of $|m_t|$
9:     Compute worst-case perturbation on selected subset:
10:     $\epsilon_{t\Omega}^* = \rho \frac{g_t(\Omega_t)}{\|g_t(\Omega_t)\|_2}$
11:     Update second-order momentum: $v_t = \beta_2 v_{t-1} + (1 - \beta_2)g_t^2$
12:     Bias correction: $\hat{m}_t = \frac{m_t}{1-\beta_1^t}, \hat{v}_t = \frac{v_t}{1-\beta_2^t}$
13:     Compute gradient at perturbed points: $g'_t = \nabla_\Theta L(\Theta_{t-1} + \epsilon_{t\Omega}^*, D_t)$
14:     Selective parameter update:
15:     **for** each parameter $i$ **do**
16:       **if** $i \in \Omega_t$ **then**
17:         $\Theta_{t,i} = \Theta_{t-1,i} - \eta \frac{\hat{m}_{t,i}}{\sqrt{\hat{v}_{t,i}+\epsilon}} g'_{t,i}$
18:       **else**
19:         $\Theta_{t,i} = \Theta_{t-1,i}$ (remain unchanged)
20:       **end if**
21:     **end for**
22: **end for**
23: **Return** $\Theta_T$

---

# D. Experimental Details

This section outlines the experimental setup used to evaluate our method. To systematically present these details, we divide this section into two parts. In Section D.1, we provide the general setup and baselines, covering datasets and task settings, baseline methods, and implementation details. In Section D.2, we describe specific analysis setups, including parameter change analysis and weight disentanglement visualization. These details ensure reproducibility and provide context for our empirical findings.

## D.1. General Setup and Baselines

### D.1.1. DATASETS AND TASK SETTINGS

**Vision Tasks.** For evaluating SAIM in the vision domain, we follow the FusionBench protocol (Tang et al., 2025a; Qiu et al., 2025), using 20 diverse image classification datasets organized into three task groups. To ensure robustness and consistency, we conduct experiments with five different random task permutations (using seeds 42-46), as shown in Table 11. For clarity, we denote the datasets by their indices: (1) SUN397(Xiao et al., 2010), (2) Stanford Cars(Krause et al., 2013), (3) RESISC45(Cheng et al., 2017), (4) EuroSAT(Helber et al., 2019), (5) SVHN(Netzer et al., 2011), (6) GTSRB(Stallkamp et al., 2012), (7) MNIST(LeCun et al., 2002), (8) DTD(Cimpoi et al., 2014), (9) Flowers102(Nilsback & Zisserman, 2008), (10) PCAM(Veeling et al., 2018), (11) FER2013(Goodfellow et al., 2013), (12) Oxford-IIIT Pets(Parkhi et al., 2012), (13) STL-10(Coates et al., 2011), (14) CIFAR-100 and (15) CIFAR-10(Krizhevsky et al., 2009), (16) Food-101(Bossard et al., 2014), (17) Fashion-MNIST(Xiao et al., 2017), (18) EMNIST(Cohen et al., 2017), (19) KMNIST(Clanuwat et al., 2018), (20) Rendered SST-2(Socher et al., 2013).

---

**Algorithm 2** Sharpness-Aware Isotropic Merging (SAIM) for Continual Learning

---

1: **Input:** Pre-trained model parameters $\theta_{pre}$, task sequence $\mathcal{T} = \{T_1, T_2, \ldots, T_N\}$, balance coefficient $\lambda$, scaling factor $\alpha$
2: **Output:** Final model parameters $\theta_{final}$
3: $\theta_0 \leftarrow \theta_{pre}$
4: **for** $t = 1, 2, 3, \ldots, N$ **do**
5:      Sharpness-Aware Fine-tuning (see Algorithm 1)
6:      $\theta_{T_t} \leftarrow \text{SA-BCD-Finetune}(\theta_{t-1}, T_t, \eta, \ldots)$
7:      $\Delta_{T_t} = \theta_{T_t} - \theta_{t-1}$
8:      $\Delta_{merged} \leftarrow \emptyset$
9:      **for** each parameter layer $k$ **do**
10:          $\Delta_{cum}^k = \theta_{t-1}^k - \theta_{pre}^k$
11:          $\Delta_{com} = \Delta_{cum}^k + \Delta_{T_t}^k$
12:          Singular value decomposition: $\Delta_{com} = U\Sigma V^\top$
13:          Adaptive singular value balancing:
14:          $\bar{\sigma} = \frac{1}{r}\sum_{i=1}^{r}\sigma_i$
15:          $\hat{\Sigma} = \bar{\sigma} + (\Sigma - \bar{\sigma}) \times \frac{1}{\sqrt{t}}$
16:          $\Delta_{merged}^k \leftarrow U\hat{\Sigma}V^\top$
17:      **end for**
18:      Update model parameters: $\theta_t \leftarrow \theta_0 + \alpha\Delta_{merged}$
19: **end for**
20: **Return** $\theta_{final} \leftarrow \theta_N$

---

- **8-task setting**: SUN397, Stanford Cars, RESISC45, EuroSAT, SVHN, GTSRB, MNIST, and DTD

- **14-task setting**: The 8-task setting plus Flowers102, PCAM, FER2013, Oxford-IIIT Pets, STL-10, and CIFAR-100

- **20-task setting**: The 14-task setting plus CIFAR-10, Food-101, Fashion-MNIST, EMNIST, KMNIST, and Rendered SST-2

These datasets span various domains including scene recognition, object classification, remote sensing, texture analysis, and character recognition. To provide a comprehensive overview, we list the detailed statistics of all downstream datasets used in our experiments in Table 10, following prior works Tang et al., 2025a; Qiu et al., 2025. The table includes the number of classes, training and test samples, and the corresponding task type for each dataset.

To ensure robustness to task ordering effects, we conduct experiments with five different random task permutations, as shown in Table 11. For all vision experiments, we use CLIP-ViT models (ViT-B/32, ViT-B/16, and ViT-L/14) as the backbone architectures.

**Language Model Tasks.** We evaluate SAIM in language domains using two benchmark suites:

**MergeBench** (He et al., 2025): This benchmark consists of five distinct domains with various tasks in each domain. We use the official *Llama-3.2-3B* and *Llama-3.1-8B* model weights in our experiments. Table 12 details the datasets and evaluation metrics for each domain.

**TRACEBench** (Wang et al., 2023b): This benchmark evaluates continual learning through a sequence of seven tasks, with each task containing 5,000 samples. We use *Llama-3.2-3B-Instruct* and *Llama-3.2-1B-Instruct* as the base model and follow the official task sequence. The evaluation metrics for each task are summarized in Table 13.

**Evaluation Protocol.** We evaluate our method in two fine-tuning settings:

- **Independent fine-tuning**: Each task is fine-tuned separately from the same pre-trained model, then merged.

- **Sequential fine-tuning**: Each task is fine-tuned from the previous merged model, forming a true continual learning pipeline.

*Table 10.* Extended downstream datasets used in our experiments.

| Dataset | #Classes | #Train (k) | #Test (k) | Task |
|---|---|---|---|---|
| SUN397 | 287 | 19.9 | 19.9 | Scene category |
| Stanford Cars | 196 | 8.1 | 8.0 | Car series |
| RESISC45 | 45 | 18.9 | 6.3 | Remote-sensing scene |
| EuroSAT | 17 | 21.6 | 2.7 | Satellite land-use |
| SVHN | 10 | 73.3 | 26.0 | Digit recognition |
| GTSRB | 43 | 39.2 | 12.6 | Traffic sign |
| MNIST | 10 | 60 | 10 | Hand-written digit |
| DTD | 47 | 3.8 | 1.9 | Texture recognition |
| Flowers102 | 102 | 1.0 | 6.1 | Flower species |
| PCAM | 2 | 262 | 32.8 | Tumour classification |
| FER2013 | 7 | 28.7 | 3.6 | Facial emotion |
| Oxford-IIIT Pets | 37 | 3.7 | 3.7 | Animal species |
| STL-10 | 10 | 5 | 8 | Object recognition |
| CIFAR-100 | 100 | 50 | 10 | Natural object |
| CIFAR-10 | 10 | 50 | 10 | Natural object |
| Food-101 | 101 | 75.8 | 25.3 | Food type |
| Fashion-MNIST | 10 | 60 | 10 | Fashion product |
| EMNIST | 10 | 60 | 10 | Hand-written digit |
| KMNIST | 10 | 60 | 10 | Kuzushiji character |
| Rendered SST-2 | 2 | 6.9 | 1.8 | Rendered sentiment |

*Table 11.* The five different task orderings used in our vision experiments.

| Order | Dataset Order by ID |
|---|---|
| 1 | $(6\to5\to7\to2\to3\to8\to1\to4\to10\to14\to13\to11\to12\to9\to20\to15\to16\to19\to18\to17)$ |
| 2 | $(7\to8\to5\to4\to2\to6\to3\to1\to13\to12\to9\to14\to10\to11\to15\to16\to17\to20\to18\to19)$ |
| 3 | $(3\to8\to2\to1\to5\to7\to6\to4\to9\to11\to13\to10\to12\to14\to15\to16\to20\to19\to18\to17)$ |
| 4 | $(4\to7\to8\to2\to1\to6\to5\to3\to11\to10\to12\to13\to14\to9\to17\to19\to18\to15\to20\to16)$ |
| 5 | $(2\to6\to4\to8\to1\to7\to5\to3\to10\to13\to9\to11\to14\to12\to17\to19\to18\to16\to20\to15)$ |

For vision tasks, we follow the FusionBench protocol proposed by (Tang et al., 2025a; Qiu et al., 2025), reporting average accuracy (ACC) and backward transfer (BWT) as defined in the main text. For MergeBench, each domain score is calculated as the average of the main metrics of all tasks within that domain (e.g., prompt-level accuracy, exact match, pass@1, etc.), and the overall score is the arithmetic mean of the five domain scores. For TRACEBench, the main metric of each of the seven tasks is extracted, and the final score is the arithmetic mean of these seven metrics.

### D.1.2. BASELINE METHODS

All baseline methods evaluated in this work are implemented following their original principles and code logic, with hyperparameters set according to the recommendations in their respective papers. Let $\theta_0$ denote the pre-trained parameters, $\theta_t$ the expert model fine-tuned for task $t$, and the task vector $\boldsymbol{\tau}_t = \theta_t - \theta_0$. After $t$ tasks, the merged model is denoted as $\theta_{\text{merged}}^{(t)}$.

**Continual Fine-tuning (C. Fine-Tuned).** The model is sequentially fine-tuned on each new task dataset $D_t$, updating parameters as $\theta^{(t)} = \text{FineTune}(\theta^{(t-1)}, D_t)$, with $\theta^{(0)} = \theta_0$. This approach adapts the model to each new task but may lead to forgetting previous knowledge.

**Stochastic Weight Averaging (SWA).** SWA incrementally averages the parameters of all task-specific models(Izmailov et al., 2018). At each step, the merged model is updated as $\theta_{\text{SWA}}^{(t)} = \frac{t-1}{t}\theta_{\text{SWA}}^{(t-1)} + \frac{1}{t}\theta_t$, which is equivalent to $\theta_{\text{SWA}}^{(T)} = \theta_0 + \frac{1}{T}\sum_{t=1}^{T}\boldsymbol{\tau}_t$. This method smooths out individual task updates and can improve robustness.

*Table 12.* Datasets used for model evaluation in MergeBench.

| Category | Dataset | Metric | # Data |
|---|---|---|---|
| Instruction-following | IFEval | Prompt level accuracy | 541 |
| Mathematics | GSM8k | EM (8-shot CoT) | 1320 |
| | MATH | EM (0-shot CoT) | 5000 |
| Multilingual understanding | M_MMLU | Accuracy | 60K |
| | M_ARC | Normalized accuracy | 10.34K |
| | M_Hellaswag | Normalized accuracy | 37.35K |
| Coding | Humaneval+ | Pass@1 | 164 |
| | MBPP+ | Pass@1 | 378 |
| Safety | WildGuardTest | RTA | 1730 |
| | HarmBench | RTA | 410 |
| | DoAnythingNow | RTA | 15.14K |
| | XSTest | Accuracy | 450 |

*Table 13.* TRACEBench tasks and evaluation metrics.

| Category | Task | Description | Metric | Avg. Len. |
|---|---|---|---|---|
| Domain-specific | ScienceQA | Science reasoning | Accuracy | 210 |
| | FOMC | Meeting summarization | Accuracy | 51 |
| | MeetingBank | Meeting QA | ROUGE-L | 2853 |
| Multi-lingual | C-STANCE | Stance detection | Accuracy | 127 |
| | 20Minuten | News summarization | SARI | 382 |
| Mathematical reasoning | NumGLUE-cm | Commonsense math | Accuracy | 32 |
| | NumGLUE-ds | Data science reasoning | Accuracy | 21 |

**Continual Task Arithmetic (C. TA).** For each task $t$, the task vector is defined as $\boldsymbol{\tau}_t = \theta_t - \theta_0$. The cumulative task vector up to step $t$ is $\boldsymbol{\tau}_{\text{cum}}^{(t)} = \sum_{i=1}^{t} \boldsymbol{\tau}_i$. The merged model is constructed by applying a scaling factor $\alpha$ to the cumulative task vector, i.e., $\theta_{\text{merged}}^{(t)} = \theta_0 + \alpha \boldsymbol{\tau}_{\text{cum}}^{(t)}$. This approach integrates knowledge from all tasks by accumulating their parameter changes(Ilharco et al., 2022).

**Continual TIES-Merging (C. TIES).** Each task vector $\boldsymbol{\tau}_t$ is pruned by keeping only the top-$k$% entries with the largest absolute values, setting the rest to zero. The pruned vector is denoted as $\text{Trim}_k(\boldsymbol{\tau}_t)$. The cumulative pruned vector is updated recursively as $\Delta_{\text{cum}}^{(t)} = \Delta_{\text{cum}}^{(t-1)} + \text{Trim}_k(\boldsymbol{\tau}_t)$, starting from $\Delta_{\text{cum}}^{(0)} = 0$. The merged model is then computed as $\theta_{\text{merged}}^{(t)} = \theta_0 + \alpha \Delta_{\text{cum}}^{(t)}$, where $\alpha$ is a scaling factor. This method emphasizes the most significant parameter changes and reduces interference(Yadav et al., 2023).

**Maximum Magnitude Selection (MagMax).** For each parameter dimension $j$, MagMax selects the update with the largest absolute value among all tasks, i.e., $\Delta\theta_{t,j}^{\text{MagMax}} = \arg\max_{s \le t} |\boldsymbol{\tau}_{s,j}|$. The merged model is given by $\theta_{\text{merged}}^{(t)} = \theta_0 + \alpha \Delta\theta_t^{\text{MagMax}}$, where $\alpha$ is a scaling coefficient. This approach aims to maximize stability by always choosing the strongest signal for each parameter(Marczak et al., 2024).

**Orthogonal Projection-based Continual Merging (OPCM).** OPCM is a projection-based scheme to mitigate task interference by enforcing orthogonality between parameter updates. Specifically, at each step, the update $\Delta\theta_t$ is projected onto the orthogonal complement of the subspace spanned by previous updates. The merged model is computed as $\theta_t^{\text{merged}} = \theta_0 + \frac{1}{\lambda_t} \left[ \lambda_{t-1} \Delta\theta_{t-1}^{\text{merged}} + \mathcal{P}^{(t-1)}(\Delta\theta_t) \right]$, where $\mathcal{P}^{(t-1)}$ denotes the orthogonal projection operator and $\lambda_t$ is a normalization factor. This approach reduces overlap between tasks and preserves unique information in the merged model (Tang et al., 2025b).

**Elastic Weight Consolidation (EWC).** EWC protects past knowledge by penalizing parameter drift weighted by diagonal Fisher information (Kirkpatrick et al., 2017). The objective is to minimize $\mathcal{L}_t(\theta) + \frac{\lambda}{2} \sum_i F_i (\theta_i - \theta_i^\star)^2$, where $\theta^\star$ is the previous optimum and $\lambda$ controls the trade-off between stability and plasticity. This regularization helps retain important parameters for previous tasks.

**Experience Replay (ER).** Current data $D_t$ is mixed with a replay buffer $\mathcal{B}$ of past samples. The optimization objective is $\min_\theta \mathbb{E}_{(x,y) \sim D_t} \mathcal{L}(\theta; x, y) + \mu \mathbb{E}_{(x,y) \sim \mathcal{B}} \mathcal{L}(\theta; x, y)$, with $\mu > 0$ controlling the replay strength. This method alleviates forgetting by revisiting previous data during training (Buzzega et al., 2020).

**O-LoRA (Wang et al., 2023a).** O-LoRA incrementally allocates task-specific LoRA subspaces and constrains the current task subspace to be orthogonal to previous ones. Denote LoRA factors of task $t$ by $(A_t, B_t)$. The orthogonality penalty is

$$\mathcal{L}_{\text{orth}} = \sum_{i<t} \|A_i^\top A_t\|_F^2, \tag{38}$$

and the training objective is $\mathcal{L}_t + \lambda \mathcal{L}_{\text{orth}}$. The effective weight update can be viewed as $W = W_0 + \sum_{i=1}^t A_i B_i$. This design reduces subspace interference without replay buffers.

**LoRAMoE (Dou et al., 2024).** LoRAMoE replaces single-adapter updates with a mixture of multiple LoRA experts and a learned router. For input representation $x$, the routed LoRA update is

$$\Delta W(x) = \frac{\alpha}{r} \sum_{e=1}^E g_e(x) B_e A_e, \tag{39}$$

with routing weights $g(x) = \text{Softmax}(W_g x)$. The layer output is $y = W_0 x + \Delta W(x) x$. By dynamic expert combination, LoRAMoE improves adaptation flexibility over single-adapter continual tuning.

**MH-MoE (Chen et al., 2026).** MH-MoE performs routing at the attention-head level instead of routing on a fully mixed post-attention representation. Let $x_h$ be the representation of head $h$. Head-wise routing computes $g_h(x_h) = \text{Softmax}(W_h x_h)$, and the MoE correction is aggregated as

$$\Delta y = \sum_{h=1}^H \sum_{e=1}^{E_h} g_{h,e}(x_h) B_{h,e} A_{h,e} x_h. \tag{40}$$

This design reduces composition collisions in routing and mitigates forgetting caused by head-mixed interference.

**PASs-MoE (Hou et al., 2026).** PASs-MoE defines a pathway activation subspace for each expert using LoRA down-projection: $\mathcal{S}_e = \text{span}(A_e^\top)$. Routing is reweighted by pathway activation strength (e.g., based on $\|A_e x\|$), and rank-level stabilization is imposed on historically important directions. Its objective is typically formulated as $\mathcal{L} = \mathcal{L}_{\text{task}} + \lambda_{\text{RS}} \mathcal{L}_{\text{rank\_stab}}$, where $\mathcal{L}_{\text{rank\_stab}}$ penalizes drift on task-critical low-rank channels. A common rank-level form is

$$\mathcal{L}_{\text{rank\_stab}} = \sum_{e=1}^E \sum_{j=1}^r I_{e,j}^{\text{hist}} \left( \|a_{e,j} - a_{e,j}^{\text{old}}\|_2^2 + \|b_{e,j} - b_{e,j}^{\text{old}}\|_2^2 \right), \tag{41}$$

where $a_{e,j}$ and $b_{e,j}$ are the current rank-$j$ LoRA factors of expert $e$, $(a_{e,j}^{\text{old}}, b_{e,j}^{\text{old}})$ are their historical references, and $I_{e,j}^{\text{hist}}$ is the historical importance weight of that rank direction. This coupling of routing alignment and rank stabilization improves retention under fixed-capacity MoE-LoRA.

**InfLoRA (Liang & Li, 2024).** InfLoRA expands a LoRA-like branch for each task while freezing the backbone and previous branches. It designs the reduction matrix $B_t$ using gradient-space information so that the update for task $t$ lies in an interference-free subspace. Let $\mathcal{N}_t$ denote the gradient subspace of the new task and $\mathcal{M}_t$ that of old tasks (estimated via DualGPM); InfLoRA enforces

$$\text{span}\{b_{t,1}, \ldots, b_{t,r}\} \subseteq \mathcal{N}_t \cap \mathcal{M}_t^\perp, \tag{42}$$

where $b_{t,i}$ are the rows of $B_t$. Only $A_t$ is tuned, keeping updates orthogonal to old-task gradients while preserving plasticity within $\mathcal{N}_t$.

**LoRA Subtraction (Liu & Chang, 2025).** LoRA-Sub addresses feature drift in exemplar-free continual learning by establishing a Drift-Resistant Space (DRS). It subtracts accumulated LoRA updates of past tasks from the pre-trained weights before processing new task data:

$$\widetilde{W}_t = W_0 - \sum_{i<t} \Delta W_i, \quad \Delta W_i = B_i A_i, \tag{43}$$

where $W_0$ is the initial pre-trained weight. The DRS is then derived from features under $\widetilde{W}_t$, and new LoRA updates are learned in this space to reduce interference without storing exemplars.

**SD-LoRA (Wu et al., 2025).** Scalable Decoupled LoRA (SD-LoRA) separates the learning of LoRA directions and magnitudes across tasks. It keeps the previously learned directions $\{A_k B_k\}_{k<t}$ fixed and only reweights them with learnable scalars $\{\alpha_k\}$ while adding a new direction for task $t$:

$$h' = \left(W_0 + \sum_{k=1}^{t} \alpha_k A_k B_k\right) x. \tag{44}$$

This decoupling enables rehearsal-free continual learning with efficient inference, since the final model directly uses the aggregated LoRA components without task-specific selection.

### D.1.3. IMPLEMENTATION DETAILS

All experiments were conducted on NVIDIA RTX 4090 (48G) GPUs using mixed precision (FP16). We utilized the AdamW optimizer with cosine learning rate scheduling and a weight decay of 0.01.

**Experiments on Vision Tasks.** For vision tasks, we used CLIP-ViT models (ViT-B/32, ViT-B/16, ViT-L/14) as the backbone architectures with input resolution 224, following the FusionBench protocol (Tang et al., 2025a; Qiu et al., 2025). The global batch size was 128 (32 for ViT-L/14), with gradient accumulation to ensure consistent effective batch size. Each ViT model was fine-tuned on each task using the AdamW optimizer with a learning rate of 1e-5. The batch size was set to 32 for ViT-L-14 and 128 for the other models, with gradient accumulation steps of 4 for ViT-L-14 and 1 for the others. The number of training epochs for each task is consistent with the ISO-C protocol (Marczak et al., 2025). For our SAIM method, we set the balance coefficient $\lambda$ to 0, the perturbation magnitude $\rho$ in SA-BCD to 0.05, and the parameter selection ratio $p$ to 0.3. All experiments use the official training splits, and no data from previous tasks is accessed during fine-tuning or merging. To evaluate robustness to task order, we repeat experiments with five different task permutations (random seeds 42 to 46) and report the mean and standard deviation (Tang et al., 2025b; Qiu et al., 2025).

**Experiments on Large Language Models.** For large language model experiments, we conducted continual learning on both the MergeBench (He et al., 2025) and TRACEBench (Wang et al., 2023b) benchmarks. For MergeBench, we used the official *Llama-3.2-3B* and *Llama-3.1-8B* weights provided by huggingface[1], and performed continual model merging across five domains (instruction, mathematics, multilingual, coding, safety) to simulate the continual learning process (He et al., 2025). In SAIM, the perturbation magnitude $\rho$ in SA-BCD was set to 0.05, the parameter selection ratio $p$ to 0.3, the scaling factor $\alpha$ to 1.0 for *Llama-3.2-3B* and 1.1 for *Llama-3.1-8B*, and the balance coefficient $\lambda$ to 0.25. For TRACEBench, we followed the TRACE and MoFO protocols (Chen et al., 2024; Wang et al., 2023b), using *Llama-3.2-3B-Instruct* and *Llama-3.2-1B-Instruct* as the base models and performing continual fine-tuning and merging according to the official task sequence (C-STANCE, FOMC, MeetingBank, ScienceQA, NumGLUE-cm, NumGLUE-ds, 20Minuten). Fine-tuning was performed with a learning rate of 2e-5, batch size of 8, and cosine decay schedule. In SAIM, the balance coefficient $\lambda$ was set to 0, with other hyperparameters the same as above, and the sequence length was set to 2048.

### D.2. Specific Analysis Setups

#### D.2.1. PARAMETER CHANGE ANALYSIS SETUP

In our parameter change analysis experiments, we used the ViT-B/32 model and analyzed parameter changes after fine-tuning on five categories of visual tasks, each containing three specific datasets:

- OCR (Optical Character Recognition): MNIST, EMNIST, KMNIST

---

[1] https://huggingface.co/MergeBench

- VQA (Visual Question Answering): STL10, CIFAR100, RenderedSST2

- Geometry (Geometric Recognition): GTSRB, RESISC45, EuroSAT

- Chart (Chart and Scene Classification): CIFAR10, Food101, SUN397

- Grounding (Object Grounding and Segmentation): Cars, OxfordIIITPet, DTD

For each task category, we performed fine-tuning using both the standard AdamW optimizer and our proposed SA-BCD optimizer, with a parameter selection ratio $p = 0.3$ and perturbation magnitude $\rho = 0.05$. All fine-tuning was conducted with a learning rate of $1e-5$ and weight decay of $0.01$. To ensure reproducibility, we set the random seed to 42 and fine-tuned separately on each of the three datasets for every task category. After fine-tuning, we computed the absolute difference between the fine-tuned and pre-trained model parameters, took the logarithm ($\log_{10}$), and plotted histograms to visualize the distribution of parameter changes under different optimizers.

### D.2.2. WEIGHT DISENTANGLEMENT VISUALIZATION SETUP

To evaluate the effectiveness of SA-BCD fine-tuning in reducing parameter interference and improving weight disentanglement, we conduct visualization experiments under two settings: (1) merging two task-specific models across two tasks, and (2) merging all eight task-specific models across two tasks.

In the first setting, the merged model is parameterized as $\theta_{\text{merge}} = \theta_0 + \alpha_1 \tau_1 + \alpha_2 \tau_2$, where $\tau_1$ and $\tau_2$ are the task vectors obtained by fine-tuning on tasks 1 and 2, respectively. In the second setting, the merged model is parameterized as $\theta_{\text{merge}} = \theta_0 + \alpha_1 \tau_1 + \alpha_2 \tau_2 + \sum_{s \notin \{1,2\}} \alpha_s \tau_s$, where $\tau_s$ denotes the task vectors for the remaining six tasks and $\alpha_s$ is obtained through grid search to maximize the average accuracy.

For both settings, we vary $(\alpha_1, \alpha_2)$ over a grid from $-0.5$ to $1.5$ with 21 evenly spaced points along each axis, resulting in a $21 \times 21$ grid. For each $(\alpha_1, \alpha_2)$ pair, we compute the disentanglement error $\xi(\alpha_1, \alpha_2)$ using the merged model obtained via SA-BCD fine-tuning. The error values are visualized using contour plots, which highlight regions in the parameter space where weight disentanglement is stronger. Since the task coefficients are real-valued, contour plots provide an effective way to illustrate the variations in loss landscape and disentanglement error across the continuous $(\alpha_1, \alpha_2)$ space.

## E. Additional Results

This section presents supplementary experimental results to further demonstrate the effectiveness of SAIM. To organize these findings systematically, we categorize them into three main parts: First, Section E.1 provides detailed performance and benchmark evaluations, including accuracy matrices and evaluations on MergeBench and TRACEBench. Second, Section E.2 covers ablation studies and training analysis, demonstrating the impact of fine-tuning strategies, parameter sensitivity, full training epochs, and full convergence. Third, Section E.3 explores model mechanisms and computational efficiency, including weight disentanglement, subspace alignment, and memory efficiency analysis.

### E.1. Detailed Performance and Benchmark Evaluations

#### E.1.1. DETAILED TASK PERFORMANCE

Table 14 extends the results from Table 1 in the main text, providing a detailed breakdown of accuracy for each of the 20 visual tasks after merging. We compare various methods (SWA, Task Arithmetic, TIES-Merging, MagMAX-IND, OPCM, and our proposed SAIM) across three CLIP-ViT backbones (ViT-B/32, ViT-B/16, ViT-L/14).

As shown in Table 14, SAIM achieves the best performance on most tasks. Particularly significant improvements are observed on complex tasks such as SUN397, RESISC45, and GTSRB, where SAIM outperforms the second-best method by approximately 5-20 percentage points. This advantage becomes more pronounced with larger models (ViT-L/14), indicating that SAIM effectively leverages the expressive capacity of larger architectures. Notably, on certain tasks like FER2013, other methods may show slight advantages, suggesting that task characteristics can influence the optimal merging strategy.

Moreover, we observe that all methods face challenges on certain tasks (e.g., EMNIST and KMNIST), likely due to their significant distribution shift from pre-training data. Even on these challenging tasks, however, SAIM demonstrates superior generalization capabilities, validating our method's effectiveness in cross-task knowledge transfer.

*Table 14.* Test set accuracy comparisons on different downstream tasks.

| Model | SUN397 | Cars | RESISC45 | EuroSAT | SVHN | GTSRB | MNIST | DTD | Flowers102 | PCAM |
|---|---|---|---|---|---|---|---|---|---|---|
| **ViT-B/32** | | | | | | | | | | |
| C. FINE-TUNED | 53.9 | 38.2 | 64.7 | **98.7** | 45.4 | 34.4 | 86.7 | 58.4 | 57.5 | 67.7 |
| AVERAGE (SWA) | 64.2 | 59.6 | 64.8 | 60.9 | 47.3 | 43.1 | 71.8 | 46.4 | 66.5 | 63.9 |
| C. TA | 62.0 | 53.7 | 60.9 | 58.1 | 48.5 | 48.9 | 79.4 | 46.1 | 61.1 | 73.4 |
| C. TIES | 62.5 | 49.1 | 55.8 | 50.9 | 54.6 | 49.3 | 82.0 | 46.7 | 58.5 | 69.9 |
| MAGMAX-IND | 63.6 | 53.1 | 59.7 | 49.1 | 53.8 | 53.1 | 79.8 | 43.2 | 56.9 | 75.1 |
| OPCM | 64.4 | 51.1 | 66.0 | 71.7 | **66.1** | 56.0 | **90.2** | 40.4 | 64.9 | 80.2 |
| **SAIM (Ours)** | **68.5** | **60.3** | **84.5** | 74.9 | 65.3 | **76.1** | 86.0 | **66.2** | **68.2** | **84.9** |
| **ViT-B/16** | | | | | | | | | | |
| C. FINE-TUNED | 62.7 | 58.0 | 67.6 | **99.1** | 46.0 | 29.2 | 93.9 | 61.9 | 64.1 | 75.2 |
| AVERAGE (SWA) | 67.1 | 64.6 | 69.3 | 63.4 | 62.4 | 52.7 | 80.7 | 46.6 | 71.8 | 63.1 |
| C. TA | 65.8 | 57.5 | 63.8 | 59.5 | 64.7 | 54.0 | 88.0 | 45.3 | 67.5 | 67.1 |
| C. TIES | 64.2 | 52.9 | 60.9 | 53.0 | 62.8 | 48.8 | 88.4 | 45.0 | 61.3 | 68.5 |
| MAGMAX-IND | 65.8 | 51.8 | 57.8 | 42.6 | 54.4 | 43.7 | 83.0 | 42.8 | 60.4 | 69.8 |
| OPCM | 67.9 | 55.9 | 73.7 | 77.5 | **74.4** | 63.2 | **94.1** | 49.2 | **72.3** | 79.6 |
| **SAIM (Ours)** | **71.9** | **71.1** | **86.5** | 85.0 | 70.4 | **84.3** | 92.3 | **70.9** | 70.8 | **80.5** |
| **ViT-L/14** | | | | | | | | | | |
| C. FINE-TUNED | 69.5 | 73.6 | 78.3 | **99.2** | 59.3 | 49.3 | **98.6** | 69.7 | 83.2 | 78.3 |
| AVERAGE (SWA) | 70.7 | 77.7 | 76.4 | 75.3 | 69.5 | 62.1 | 93.7 | 57.7 | 80.0 | 73.6 |
| C. TA | 70.4 | 74.1 | 73.9 | 66.3 | 69.9 | 65.6 | 95.1 | 56.6 | 78.6 | 70.4 |
| C. TIES | 69.7 | 70.3 | 65.3 | 47.9 | 76.1 | 63.6 | 94.7 | 54.4 | 77.9 | 72.3 |
| MAGMAX-IND | 73.1 | 73.7 | 75.6 | 64.6 | 73.7 | 68.8 | 94.6 | 56.1 | 78.0 | 71.7 |
| OPCM | 73.1 | 78.3 | 82.4 | 80.2 | 80.8 | 80.4 | 97.4 | 61.6 | **84.8** | 76.3 |
| **SAIM (Ours)** | **77.8** | **85.8** | **93.9** | 92.6 | **86.2** | **91.3** | 97.3 | **78.1** | 82.9 | **90.3** |

| Model | FER2013 | OxfordIIITPet | STL10 | CIFAR100 | CIFAR10 | Food101 | FashionMNIST | EMNIST | KMNIST | RenderedSST2 |
|---|---|---|---|---|---|---|---|---|---|---|
| **ViT-B/32** | | | | | | | | | | |
| C. FINE-TUNED | **58.3** | 68.5 | 86.7 | 40.2 | 70.5 | 50.0 | **90.7** | 72.4 | **54.5** | 54.5 |
| AVERAGE (SWA) | 50.2 | **84.1** | **97.0** | **69.8** | 92.7 | **80.4** | 71.3 | 15.0 | 11.5 | 61.8 |
| C. TA | 51.4 | 82.3 | 94.9 | 64.6 | 91.4 | 71.9 | 73.9 | 17.8 | 12.2 | 59.9 |
| C. TIES | 49.5 | 81.3 | 95.2 | 63.7 | 91.2 | 70.2 | 73.7 | 17.8 | 16.9 | 59.8 |
| MAGMAX-IND | 56.5 | 79.9 | 94.6 | 68.7 | 91.9 | 73.8 | 74.3 | 18.3 | 15.4 | 63.9 |
| OPCM | 55.8 | 82.9 | 95.9 | 67.6 | **92.8** | 74.0 | 76.3 | 22.4 | 18.3 | 64.6 |
| **SAIM (Ours)** | 49.9 | **84.1** | 95.5 | 69.6 | 92.5 | 78.2 | 81.2 | **74.3** | 42.3 | **69.3** |
| **ViT-B/16** | | | | | | | | | | |
| C. FINE-TUNED | **60.5** | 84.5 | 90.5 | 38.8 | 73.6 | 61.9 | **89.7** | 83.3 | **51.5** | 72.8 |
| AVERAGE (SWA) | 50.9 | 89.6 | **98.0** | 72.9 | 94.2 | **85.9** | 73.3 | 15.6 | 12.4 | 62.5 |
| C. TA | 50.7 | 89.3 | 97.0 | 68.0 | 93.1 | 80.3 | 75.7 | 18.1 | 16.7 | 61.8 |
| C. TIES | 50.4 | 87.9 | 96.3 | 63.1 | 91.7 | 78.0 | 75.0 | 23.4 | 24.9 | 61.5 |
| MAGMAX-IND | 57.7 | 88.8 | 97.5 | 71.5 | 94.4 | 81.3 | 77.2 | 24.5 | 25.0 | 59.4 |
| OPCM | 59.5 | **91.8** | 97.7 | 73.2 | 94.7 | 83.1 | 81.3 | 26.5 | 23.4 | 66.8 |
| **SAIM (Ours)** | 49.7 | 90.1 | 97.8 | **76.1** | **95.0** | 85.3 | 86.2 | **92.9** | 49.9 | **74.2** |
| **ViT-L/14** | | | | | | | | | | |
| C. FINE-TUNED | **68.0** | 92.1 | 94.5 | 60.5 | 85.7 | 74.8 | **93.1** | 89.0 | 59.2 | 78.8 |
| AVERAGE (SWA) | 52.7 | 94.2 | 99.2 | 81.7 | 97.0 | 90.7 | 77.4 | 16.1 | 10.4 | 66.1 |
| C. TA | 55.7 | 94.2 | 98.6 | 79.1 | 91.6 | 87.6 | 80.8 | 17.6 | 10.6 | 63.6 |
| C. TIES | 57.6 | 93.5 | 97.8 | 74.0 | 95.6 | 84.7 | 79.7 | 20.2 | 12.6 | 58.4 |
| MAGMAX-IND | 52.9 | 93.9 | 98.7 | 82.1 | 97.3 | 89.5 | 81.6 | 19.2 | 11.1 | 68.4 |
| OPCM | 61.8 | **95.4** | 99.2 | 83.0 | 97.8 | **90.9** | 86.0 | 26.4 | 14.7 | 71.0 |
| **SAIM (Ours)** | 53.5 | 95.2 | **99.6** | **85.4** | **98.3** | 90.2 | 90.7 | **95.4** | **82.4** | **82.5** |

### E.1.2. ACCURACY MATRICES ACROSS TASK SEQUENCES

Figure 4 shows the accuracy matrices of SAIM under different model sizes and task numbers, helping visualize how well knowledge is retained during continual learning. In each matrix, rows indicate the order in which tasks are merged, and columns show the test accuracy for each task. Diagonal entries represent the accuracy on the current task, while off-diagonal entries reflect how well the model remembers previous tasks and transfers knowledge.

From these matrices, we observe several clear trends. First, as the model size increases from ViT-B/32 to ViT-L/14, the overall accuracy improves and the values become more uniformly high (see subfigures a-c). Second, when the number of tasks increases from 8 to 20, SAIM remains stable, keeping high accuracy even for earlier tasks in longer sequences (see subfigures d-f), with most values above 80% in the ViT-L/14 20-task matrix.

SAIM also performs well when switching between very different tasks. For example, in the ViT-L/14 results, the model maintains high accuracy when moving from scene recognition (SUN397) to digit recognition (MNIST) and then to remote sensing (RESISC45). The consistently bright colors in the matrices indicate that SAIM can integrate knowledge from diverse domains with little interference between tasks.

### E.1.3. RESULTS OF FIVE-TASK MODEL MERGING ON MERGEBENCH

In the main text, we primarily discuss results on the *Llama-3.2-3B* scale. Here, we provide the corresponding evaluation for the *Llama-3.1-8B* model. Table 15 presents performance across four domains, excluding the Math task[2]. Despite this, SAIM achieves the highest average score (0.494) against all baselines. For completeness, Table 16 provides results including all five task models. Even with the inclusion of the outlier Math task, SAIM maintains the highest average score (0.551), demonstrating its robustness across diverse task domains.

*Table 15.* Comparison of Model Merging Methods on MergeBench with *Llama-3.1-8B* (Excluding Math).

| Method | Task Performance (↑) | | | | Avg. (↑) |
|---|---|---|---|---|---|
| | Instr. | Multi. | Coding | Safety | |
| Pre-Trained | 0.080 | 0.532 | 0.363 | 0.370 | 0.336 |
| Fine-Tuned | 0.375 | 0.540 | 0.563 | 0.774 | 0.563 |
| TIES-Merging | 0.062 | 0.487 | 0.371 | 0.789 | 0.427 |
| DARE | 0.201 | 0.321 | 0.439 | 0.760 | 0.430 |
| SWA | 0.028 | **0.531** | 0.468 | 0.758 | 0.446 |
| MagMAX-IND | 0.125 | 0.408 | 0.511 | **0.841** | 0.471 |
| Task Arithmetic | **0.205** | 0.377 | 0.519 | 0.802 | 0.476 |
| **SAIM (Ours)** | 0.204 | 0.471 | **0.520** | 0.779 | **0.494** |

*Table 16.* Comparison of Model Merging Methods on MergeBench with *Llama-3.1-8B* (Including Math).

| Method | Task Performance (↑) | | | | | Avg. (↑) |
|---|---|---|---|---|---|---|
| | Instr. | Math | Multi. | Coding | Safety | |
| Pre-Trained | 0.080 | 0.563 | 0.532 | 0.363 | 0.370 | 0.382 |
| Fine-Tuned | 0.375 | 0.403 | 0.540 | 0.563 | 0.774 | 0.531 |
| SWA | 0.079 | 0.726 | **0.528** | **0.497** | 0.573 | 0.480 |
| TIES-Merging | 0.062 | 0.777 | 0.519 | 0.465 | 0.635 | 0.492 |
| MagMAX | 0.209 | 0.785 | 0.465 | 0.453 | 0.612 | 0.505 |
| DARE | 0.208 | 0.743 | 0.473 | 0.474 | 0.594 | 0.498 |
| Task Arithmetic | 0.195 | 0.772 | 0.489 | 0.492 | 0.587 | 0.507 |
| **SAIM (Ours)** | **0.326** | **0.796** | 0.464 | 0.475 | **0.694** | **0.551** |

### E.1.4. EVALUATION ON TRACEBENCH WITH *Llama-3.2-3B-Instruct*

We also evaluated our method using the larger *Llama-3.2-3B-Instruct* model on TRACEBench. Table 17 presents these results, which exhibit similar superior performance as observed with the 1B model in the main text. Specifically, SAIM achieves the highest average score (0.547), outperforming competitive baselines like DARE (0.491). This consistency highlights the scalability and robust generalization capability of our SA-BCD optimization approach across different scale architectures.

## E.2. Ablation Studies and Training Analysis

### E.2.1. ANALYSIS OF FINE-TUNING STRATEGIES

To explore the impact of different fine-tuning paradigms on merging performance, we compare independent fine-tuning (each task starts from the pre-trained model) and sequential fine-tuning (each task starts from the previously merged model). The experimental settings follow Section 4.1, with results averaged over five task orders. As shown in Table 18, sequential fine-tuning significantly improves accuracy for all methods, with gains exceeding 10% in some cases. This can be attributed

---

[2]Math is excluded in our primary 8B analysis because the official checkpoint exhibits a significant distribution mismatch with the evaluation sets, resulting in lower performance than the pre-trained base (0.403 vs. 0.563).

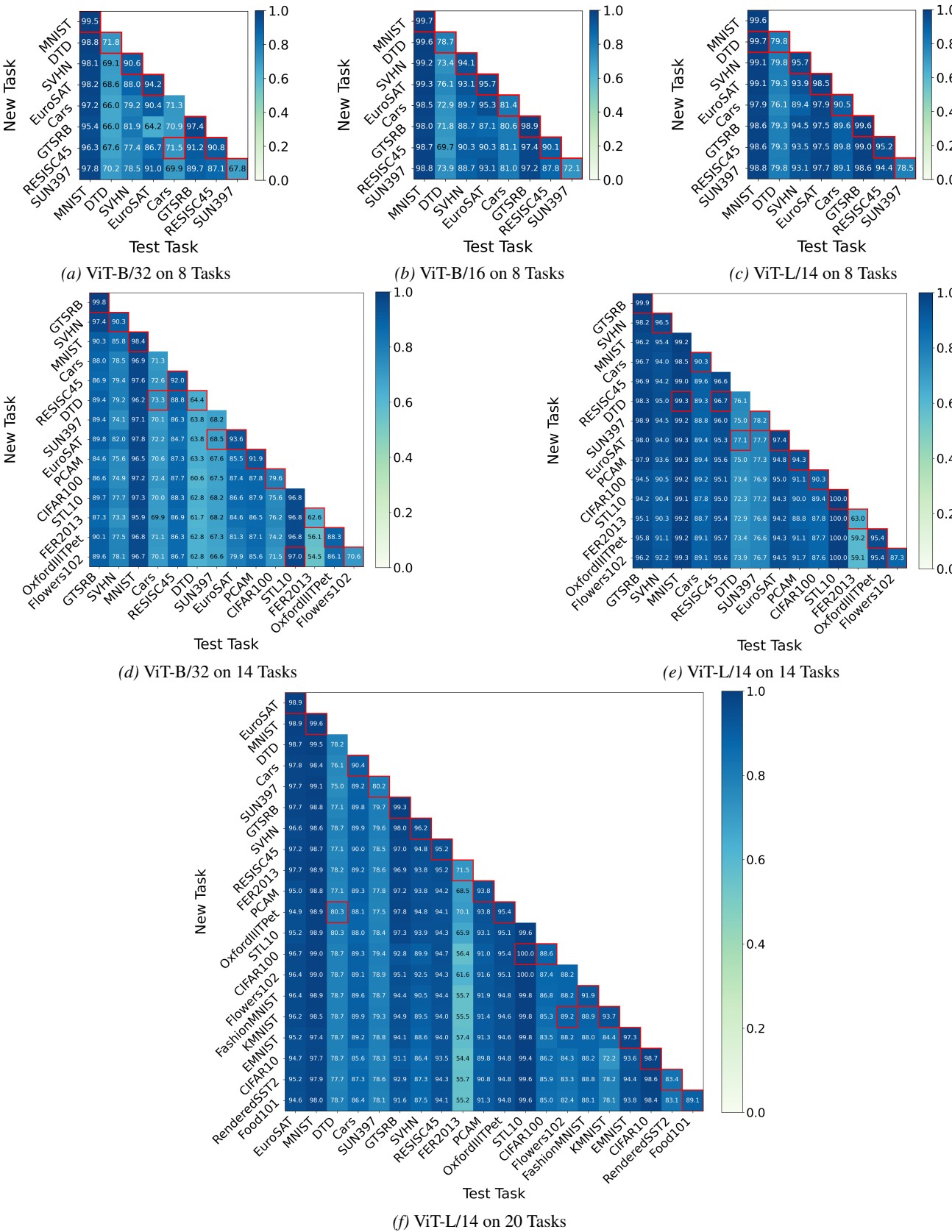

*Figure 4.* Accuracy matrices of SAIM for different model scales and task settings: (a)-(c) show 8 tasks, (d)-(e) 14 tasks, and (f) 20 tasks. Rows indicate new tasks, columns indicate test tasks.

*Table 17.* Comparison of Model Merging Methods on TRACEBench with *Llama-3.2-3B-Instruct.*

| Method | Task Performance (↑) | | | | | | | Avg. (↑) | BWT (↑) |
|---|---|---|---|---|---|---|---|---|---|
| | C-STANCE | FOMC | MeetingBank | ScienceQA | NumGLUE-cm | NumGLUE-ds | 20Minuten | | |
| Pre-Trained | 0.408 | 0.353 | 0.205 | 0.896 | 0.171 | 0.220 | 0.386 | 0.377 | - |
| Fine-Tuned | 0.542 | 0.684 | 0.432 | 0.934 | 0.610 | 0.646 | 0.390 | 0.606 | - |
| SWA | 0.462 | 0.567 | 0.221 | 0.914 | 0.439 | 0.415 | 0.389 | 0.487 | -0.025 |
| Task Arithmetic | 0.469 | 0.561 | 0.219 | 0.910 | 0.463 | 0.402 | 0.386 | 0.487 | -0.004 |
| MagMAX-IND | **0.482** | 0.520 | 0.234 | 0.906 | 0.463 | 0.421 | 0.391 | 0.488 | -0.031 |
| TIES-Merging | 0.467 | 0.571 | 0.222 | 0.913 | 0.463 | 0.402 | 0.391 | 0.490 | -0.003 |
| DARE | 0.463 | **0.587** | 0.220 | 0.906 | 0.463 | 0.409 | 0.387 | 0.491 | **0.011** |
| **SAIM (Ours)** | 0.475 | 0.569 | **0.249** | **0.924** | **0.610** | **0.610** | 0.391 | **0.547** | -0.019 |

to: (1) more consistent update directions across tasks, reducing sign conflicts and interference during merging (Marczak et al., 2024); and (2) enhanced knowledge transfer, allowing the model to accumulate multi-task information more effectively. SAIM consistently achieves superior performance under both strategies, demonstrating its robustness.

*Table 18.* Comparison of independent (Ind) and sequential (Seq) fine-tuning for SAIM, Task Arithmetic, and TIES-Merging on ViT-B/32 and ViT-B/16.

| Method | FT | ViT-B/32 | | | ViT-B/16 | | |
|---|---|---|---|---|---|---|---|
| | | 8 tasks | 14 tasks | 20 tasks | 8 tasks | 14 tasks | 20 tasks |
| C. Task Arithmetic | Ind | $67.5_{\pm0.0}$ | $66.5_{\pm0.0}$ | $60.0_{\pm0.0}$ | $77.1_{\pm0.0}$ | $70.9_{\pm0.0}$ | $64.2_{\pm0.0}$ |
| | Seq | $77.8_{\pm0.7}$ | $75.9_{\pm0.6}$ | $71.5_{\pm1.0}$ | $81.0_{\pm0.3}$ | $78.2_{\pm0.7}$ | $76.3_{\pm0.6}$ |
| C. TIES-Merging | Ind | $49.0_{\pm10.2}$ | $66.2_{\pm0.6}$ | $59.9_{\pm0.7}$ | $66.8_{\pm3.7}$ | $70.5_{\pm0.8}$ | $63.0_{\pm1.6}$ |
| | Seq | $78.6_{\pm0.6}$ | $74.9_{\pm0.9}$ | $71.8_{\pm0.5}$ | $81.7_{\pm0.9}$ | $79.1_{\pm0.6}$ | $77.1_{\pm0.7}$ |
| **SAIM (Ours)** | Ind | $82.1_{\pm0.7}$ | $\mathbf{78.7}_{\pm0.5}$ | $73.6_{\pm0.3}$ | $86.2_{\pm0.5}$ | $82.0_{\pm0.5}$ | $79.0_{\pm0.9}$ |
| | Seq | $\mathbf{82.5}_{\pm1.4}$ | $\mathbf{78.7}_{\pm1.3}$ | $\mathbf{76.4}_{\pm2.0}$ | $\mathbf{87.9}_{\pm0.6}$ | $\mathbf{83.9}_{\pm1.0}$ | $\mathbf{81.9}_{\pm1.0}$ |

### E.2.2. PARAMETER SENSITIVITY ANALYSIS

To comprehensively evaluate the robustness of SAIM and determine optimal hyperparameter configurations, we systematically analyze three key hyperparameters: the balance factor $\lambda$, parameter selection ratio $p$, and perturbation magnitude $\rho$. All experiments are conducted on the ViT-B/32 backbone, considering three settings with 8, 14, and 20 tasks. For each setting, we repeat experiments with five different task orders to ensure reliability and stability.

**Balance Factor $\lambda$ Analysis:** As shown in Figure 5(a), we perform a detailed grid search for $\lambda$ in the range $[-0.5, 0.5]$. The balance factor $\lambda$ controls the trade-off between historical knowledge and new task updates, which is crucial for continual learning. Results indicate that for all task numbers, the model achieves peak average accuracy when $\lambda$ is near zero. Specifically, accuracy reaches approximately 83% for 8 tasks, 78% for 14 tasks, and 73% for 20 tasks when $\lambda \approx 0$. Performance drops significantly as $\lambda$ deviates towards either extreme, highlighting the importance of balancing historical and new knowledge. Notably, the optimal $\lambda$ remains stable near zero as the number of tasks increases, demonstrating good cross-task robustness.

**Parameter Selection Ratio $p$ Analysis:** We search $p$ in the range $[0.1, 1.0]$, which determines the proportion of parameters updated in each SA-BCD iteration. As shown in Figure 5(b), the model achieves optimal performance when $p \approx 0.3$. Smaller $p$ values (e.g., 0.1) result in insufficient updates, limiting adaptation to new tasks, while larger $p$ values (e.g., 0.8 or 1.0) cause excessive parameter changes, disrupting existing knowledge and increasing task interference. Especially as the number of tasks grows, moderate sparsity ($p \approx 0.3$) becomes more important for reducing interference and maintaining generalization. This finding aligns with prior works such as MoFO and DARE, which emphasize the critical role of update sparsity in mitigating catastrophic forgetting.

**Perturbation Magnitude $\rho$ Analysis:** We investigate the effect of $\rho$ in $[0, 0.005, 0.05, 0.1, 0.5, 1.5]$, which controls the gradient perturbation size in SA-BCD. As shown in Figure 5(c), moderate perturbation ($\rho \approx 0.05$) significantly improves

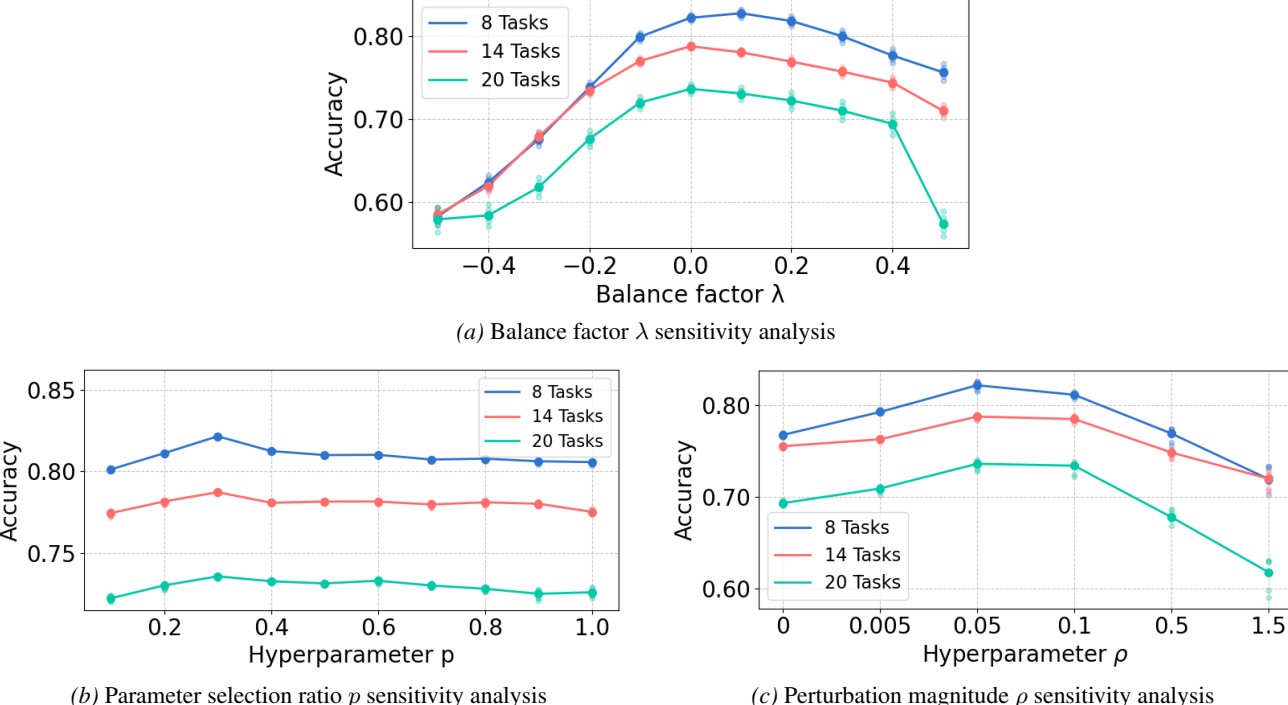

*(a)* Balance factor $\lambda$ sensitivity analysis

*(b)* Parameter selection ratio $p$ sensitivity analysis

*(c)* Perturbation magnitude $\rho$ sensitivity analysis

*Figure 5.* Parameter sensitivity analysis for SAIM on ViT-B/32 with 8, 14, and 20 tasks: (a) balance factor $\lambda$, (b) parameter selection ratio $p$, and (c) perturbation magnitude $\rho$. All results are averaged over five different task orders.

model performance. Too small values (e.g., 0 or 0.005) hinder exploration of flat regions in the loss landscape, while too large values (e.g., 0.5) destabilize optimization. The impact of $\rho$ is more pronounced with 20 tasks, indicating that as task complexity increases, guiding the model towards flatter minima becomes increasingly important. This supports our theoretical analysis that flat minima help reduce task interference and enhance knowledge retention in continual learning.

### E.2.3. IMPACT OF FULL TRAINING EPOCHS ON *Llama-3.2-1B-Instruct*

To ensure a fair computational comparison with baselines, our main experiments utilize halved training epochs. We further evaluate SAIM using full training epochs as a reference. Table19 presents the performance results, and Table20 provides the fine-tuning duration for each task.

*Table 19.* Performance comparison of SAIM on TRACEBench with *Llama-3.2-1B-Instruct* using **Halved Epochs** vs. **Full Epochs**.

| Method | Task Performance ($\uparrow$) | | | | | | | Avg. ($\uparrow$) | BWT ($\uparrow$) |
|---|---|---|---|---|---|---|---|---|---|
| | C-STANCE | FOMC | MeetingBank | ScienceQA | NumGLUE-cm | NumGLUE-ds | 20Minuten | | |
| Pre-Trained | 0.339 | 0.258 | 0.204 | 0.678 | 0.122 | 0.165 | 0.380 | 0.306 | - |
| Fine-Tuned | 0.498 | 0.599 | 0.371 | 0.852 | 0.390 | 0.579 | 0.388 | 0.525 | - |
| SAIM (Halved Epochs) | 0.466 | **0.518** | 0.259 | 0.766 | **0.488** | 0.476 | 0.384 | 0.480 | **-0.022** |
| SAIM (Full Epochs) | **0.471** | 0.506 | **0.266** | **0.812** | 0.435 | **0.483** | **0.402** | **0.482** | -0.052 |

### E.2.4. IMPACT OF FULL CONVERGENCE ON VISION TASKS

To evaluate the influence of convergence and computational budget on our method's relative advantage, we extend the training epochs to ensure full convergence (AdamW optimized models reach near-zero training loss and stabilized validation accuracy) on the ViT-B/32 architecture for the sequence of six tasks. As shown in Table 21, SA-BCD consistently outperforms the AdamW baseline even under full convergence conditions. SA-BCD achieves an average accuracy of 80.90%, surpassing AdamW's 72.95% by a significant margin. This observation confirms that the advantages of SA-BCD are fundamental to its optimization trajectory and conflict-resolution capabilities, rather than stemming merely from being

*Table 20.* Comparison of fine-tuning time (seconds) for SAIM on TRACEBench with *Llama-3.2-1B-Instruct* using **Halved Epochs** vs. **Full Epochs**.

| Setting | Task Fine-tuning Time (s) | | | | | | | Avg. ($\downarrow$) |
|---|---|---|---|---|---|---|---|---|
| | C-STANCE | FOMC | MeetingBank | ScienceQA | NumGLUE-cm | NumGLUE-ds | 20Minuten | |
| Halved Epochs | 2298.19 | 1380.55 | 3211.08 | 1381.32 | 2304.44 | 2306.22 | 3225.53 | 2301.05 |
| Full Epochs | 4423.29 | 2658.21 | 6172.68 | 2655.37 | 4434.18 | 4428.12 | 6185.02 | 4422.41 (1.92×) |

early in the training process. The ability of SA-BCD to navigate optimization conflicts yields structurally superior minima that inherently benefit sequential merging.

*Table 21.* Performance comparison under full convergence on ViT-B/32.

| Method | Convergence Epochs | | | | | | Avg. ACC (%) $\uparrow$ |
|---|---|---|---|---|---|---|---|
| | GTSRB | SVHN | MNIST | Stanford | RESISC45 | DTD | |
| AdamW | 34 | 39 | 36 | 46 | 32 | 30 | 72.95 |
| **SA-BCD (Ours)** | 34 | 43 | 26 | 39 | 46 | 50 | **80.90** |

## E.3. Model Mechanisms and Computational Efficiency

### E.3.1. MULTI-TASK WEIGHT DISENTANGLEMENT AND SUBSPACE ALIGNMENT

**Multi-task Weight Disentanglement Analysis.** To extend our analysis beyond the two-task scenario presented in the main text, we evaluate weight disentanglement in complex multi-task settings. For multi-task scenarios, the disentanglement error is defined as:

$$\xi_{\text{all}}(\alpha_1, \alpha_2) = \sum_{t=1}^{2} \mathbb{E}_{\boldsymbol{x} \in X^{(t)}} \left[ \text{dist}\big(f(\boldsymbol{x}; \boldsymbol{\theta}_t^{ref}), \ f(\boldsymbol{x}; \boldsymbol{\theta}_{merged})\big) \right] \tag{45}$$

where $\boldsymbol{\theta}_{merged} = \boldsymbol{\theta}_0 + \alpha_1 \boldsymbol{\tau}_1 + \alpha_2 \boldsymbol{\tau}_2 + \sum_{s \notin \{1,2\}} \alpha_s \boldsymbol{\tau}_s$ and $\boldsymbol{\theta}_t^{ref} = \boldsymbol{\theta}_0 + \alpha_t \boldsymbol{\tau}_t + \sum_{s \notin \{1,2\}} \alpha_s \boldsymbol{\tau}_s$.

The visualization of $\xi_{\text{all}}$ under different $\alpha_1$ and $\alpha_2$ (see Fig. 6) shows that the merged model under SA-BCD fine-tuning achieves lower disentanglement errors in the multi-task scenario, further confirming that our method enables the merged model to better reflect the independent contributions of each task even in more complex settings.

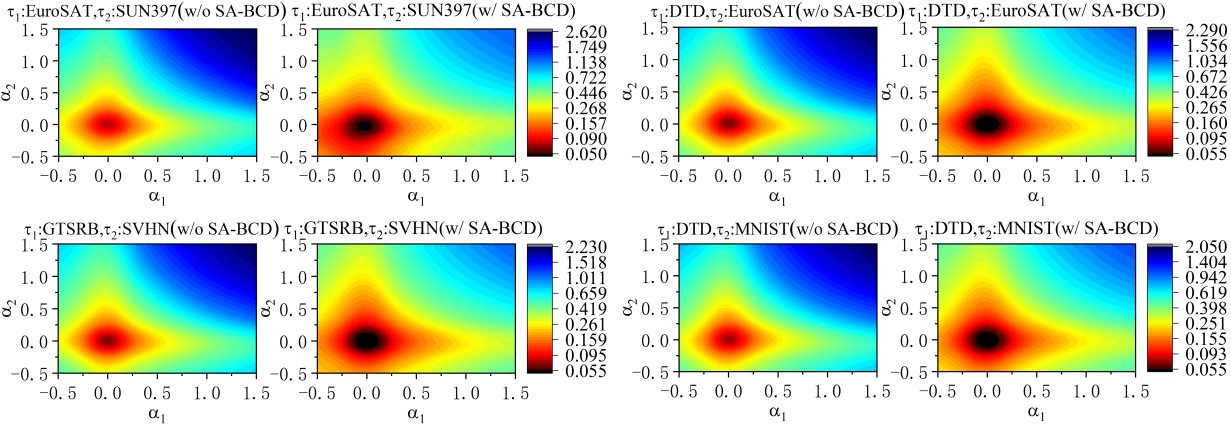

*Figure 6.* Visualization of weight disentanglement error in the eight-task merging scenario. Only two task coefficients are adjusted while the others are fixed, demonstrating the weight disentanglement capability of the merged model in a multi-task setting.

**Subspace Alignment with Increasing Number of Tasks.** ISO-C (Marczak et al., 2025) established that the subspace alignment ratio (SAR) between the merged model and each task model correlates strongly with merging performance. SAR quantifies the overlap between the task-specific parameter update matrix and the principal subspace of the merged

model. We plot the curves of average SAR and test accuracy as the number of tasks increases (see Fig. 7). The results show that the SAIM method generally maintains higher SAR and accuracy compared to baselines as the number of tasks increases, indicating better subspace alignment between the merged model and each task model, thereby enhancing multi-task knowledge integration and overall performance.

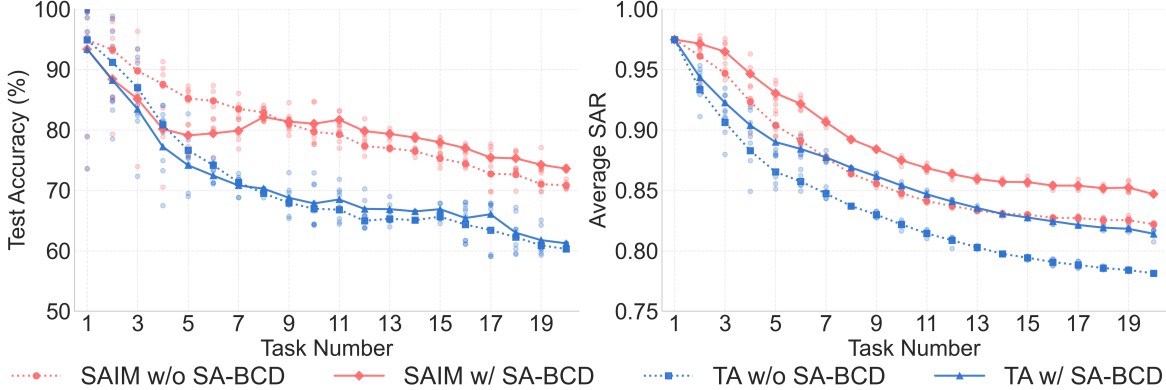

*Figure 7.* Average subspace alignment ratio (SAR) and test accuracy versus the number of tasks on ViT-B/32. Results are averaged over five different task orders. The left panel shows the evolution of test accuracy during continual learning for different methods, while the right panel shows the change in average SAR. The SAIM method generally achieves higher accuracy and SAR as the number of tasks increases, indicating enhanced multi-task knowledge integration and subspace alignment.

### E.3.2. MEMORY AND COMPUTATIONAL EFFICIENCY ANALYSIS

We evaluate the peak GPU memory and average training time using the ViT-B/32 architecture across a sequence of six tasks, following Order 1. As reported in Table 22, SAIM incurs an 8.5% increase in peak memory overhead (+624 MB) and a 60% increase in training time compared to OPCM, primarily due to the dual backward passes required by SA-BCD. However, this localized computational investment effectively mitigates catastrophic parameter interference. While the performance improvement is already notable on a six-task sequence (82.74% for SAIM vs. 79.30% for OPCM), this advantage becomes increasingly pronounced as the sequence length grows. As demonstrated in Table 1 of the main paper, SAIM significantly outperforms baselines as the task scale expands, while strictly maintaining its rehearsal-free nature.

*Table 22.* Memory, Time, and Performance Comparison on ViT-B/32 (6 tasks).

| Method | Finetune Mem (MB) ↓ | Merge Mem (MB) ↓ | Avg. Train Time (s) ↓ | Avg. Acc (%) ↑ |
|---|---|---|---|---|
| ER | 7614.75 | - | 948.34 | 81.67 |
| EWC | 9352.61 | - | 846.31 | 84.23 |
| MagMax-IND | 7288.27 | 2358.41 | 767.09 | 76.36 |
| OPCM | 7290.52 | 3364.10 | 785.94 | 79.30 |
| **SAIM (Ours)** | **7914.45** | **2456.69** | **1260.83** | **82.74** |

