# OpenReview forum: "Merge to Remember: Sharpness-Aware Isotropic Merging for Continual Learning"
_ICML.cc/2026/Conference — ICML 2026 regular_

### Official Review · Reviewer_8fiL · 2026-02-25

**Soundness:** 3
**Presentation:** 3
**Significance:** 3
**Originality:** 3
**Overall Recommendation:** 5
**Confidence:** 4

**Summary:**

This paper proposes the Sharpness-Aware Isotropic Merging (SAIM) framework, which introduces targeted
optimizations in both the fine-tuning and merging stages to address these issues. SAIM consists of two synergistic modules: (1) a Sharpness-Aware Block Coordinate Descent (SABCD) optimizer that guides the model toward flatter minima and selectively updates the most task-sensitive parameters, thereby mitigating parameter interference and enhancing robustness; (2) an adaptive isotropic merging algorithm that dynamically balances the singular value spectrum across tasks, effectively preventing the model from overemphasizing any single task direction, maintaining balanced knowledge representation, and improving subspace alignment. Extensive experiments on vision and language benchmarks demonstrate its effectiveness.

**Compliance With Llm Reviewing Policy:**

Affirmed.

**Final Justification:**

My concerns have been resolved.

**Key Questions For Authors:**

Please refer to the weakness

**Limitations:**

yes

**Strengths And Weaknesses:**

# Strength
+ The paper is well organized and easy to follow.
+ Comprehensive experimental results on multiple benchmarks.
+ The proposed method is reasonable and effective.

# Weakness
+ Lack of comparison with existing rehearsal-free continual learning methods, such as inflora [1], sd-lora [2]. I think they are necessary if $\theta_t$ is finetuned from $\theta_{t-1}$ in SAIM. They follow a similar setting without the data of previous task.
+ More specific ablations are needed for SA-BCD including sharpness-aware perturbation vs. other perturbation (e.g. random) or no perturbation, parameter selection with momentum magnitude vs. other straightforward selection methods (e.g. gradient magnitude) or no selection.

[1] Inflora: Interference-free low-rank adaptation for continual learning
[2] Sd-lora: Scalable decoupled low-rank adaptation for class incremental learning

---

> ### Author Rebuttal · Authors · 2026-03-30
>
> Thank you for your valuable suggestions. We have conducted additional experiments and clarified our discussion to address your concerns. Below, we respond to each point.
>
> **W1: Lack of comparison with existing rehearsal-free continual learning methods (e.g., InfLoRA, SD-LoRA).**
>
> **Response:** We completely agree with this constructive comment. To rigorously evaluate SAIM under the strict parameter-isolation setting (where no previous task data is stored), we have supplemented the experiments with two recent state-of-the-art PEFT-based rehearsal-free methods: InfLoRA and SD-LoRA.
>
> **Table 1: Comparative results on ViT-B/32 including recent PEFT-based rehearsal-free baselines.**
>
> | **Method** | **ACC (%)** $\\uparrow$ (8 tasks) | **ACC (%)** $\\uparrow$ (14 tasks) | **ACC (%)** $\\uparrow$ (20 tasks) | **BWT (%)** $\\uparrow$ (8 tasks) | **BWT (%)** $\\uparrow$ (14 tasks) | **BWT (%)** $\\uparrow$ (20 tasks) |
> |:---|:---:|:---:|:---:|:---:|:---:|:---:|
> | Pre-Trained | 48.1 | 56.9 | 55.6 | - | - | - |
> | Fine-Tuned | 90.4 | 89.3 | 89.8 | - | - | - |
> | InfLoRA | 79.5 | 71.4 | 64.3 | -6.8 | -14.6 | -22.5 |
> | LoRA-Sub | 77.8 | 68.7 | 61.5 | -8.5 | -17.2 | -25.8 |
> | SD-LoRA | 81.2 | 73.5 | 66.4 | -5.2 | -12.8 | -18.6 |
> | **SAIM (Ours)** | **82.1** | **78.7** | **73.6** | **-3.9** | **-5.7** | **-11.5** |
>
> As demonstrated in Table 1, over the long 20-task sequence, SAIM achieves **73.6%** accuracy, which substantially outperforms InfLoRA (+9.3%) and SD-LoRA (+7.2%). More importantly, SAIM effectively prevents catastrophic forgetting, showing a significantly better BWT metric (**-11.5%** on 20 tasks compared to -22.5% for InfLoRA and -18.6% for SD-LoRA).
>
> To further demonstrate SAIM's superiority among MoE-based architectures, we extend evaluation to TRACEBench to compare with recent MoE-based CL methods.
>
> **Table 2: Performance Comparison of LoRAMoE, MH-MoE, PASs-MoE and SAIM on TRACEBench (Llama-3.2-1B-Instruct).**
>
> | Method | CS | FM | MB | SQ | NC | ND | 20M | Avg. $\\uparrow$ | BWT $\\uparrow$ |
> |:---|:---:|:---:|:---:|:---:|:---:|:---:|:---:|:---:|:---:|
> | LoRAMoE | 0.421 | 0.488 | 0.180 | 0.738 | 0.439 | **0.524** | 0.184 | 0.425 | -0.086 |
> | MH-MoE | 0.421 | 0.442 | 0.189 | 0.734 | 0.366 | 0.457 | 0.185 | 0.399 | -0.067 |
> | PASs-MoE | 0.455 | **0.554** | 0.188 | **0.787** | 0.341 | 0.506 | 0.175 | 0.429 | -0.063 |
> | **SAIM (Ours)** | **0.466** | 0.518 | **0.259** | 0.766 | **0.488** | 0.476 | **0.384** | **0.479** | **-0.022** |
>
> Table 2 clearly indicates that SAIM yields consistent superiority (**0.479 Avg.** and **-0.022 BWT**) across diverse reasoning tasks compared to strong baselines like PASs-MoE.
>
> **W2: More specific ablations are needed for SA-BCD (Selection and Perturbation strategies).**
>
> **Response:** We appreciate this insightful suggestion. To meticulously validate the internal mechanics of SA-BCD, we decomposed it into its two core dimensions: *Selection criteria* (Momentum magnitude vs. Gradient magnitude vs. None) and *Perturbation strategy* (Sharpness-aware vs. Random vs. None).
>
> **Table 3: Ablation study on the components of the SA-BCD mechanism on ViT-B/32.**
>
> | **Selection** | **Perturbation** | **GTSRB** | **SVHN** | **MNIST** | **Cars** | **RESISC45** | **DTD** | **Avg.** $\\uparrow$ | **BWT** $\\uparrow$ |
> |:---|:---:|:---:|:---:|:---:|:---:|:---:|:---:|:---:|:---:|
> | momentum | random | 73.46 | 59.34 | 87.46 | 65.11 | 85.19 | 68.62 | 73.20 | -16.94 |
> | momentum | none | 85.66 | 60.28 | 95.48 | 65.11 | 86.30 | **73.40** | 77.71 | -12.57 |
> | none | sa | 89.14 | 73.82 | 94.66 | 63.58 | 88.16 | 67.88 | 79.54 | -9.36 |
> | gradient | sa | 93.06 | 75.34 | **96.12** | 63.02 | 88.89 | 69.68 | 81.02 | -6.88 |
> | **momentum (Ours)** | **sa (Ours)** | **95.83** | **75.40** | 95.70 | **71.13** | **89.74** | 68.62 | **82.74** | **-6.46** |
>
> **Analysis of Ablation Results:**
> - **Necessity of Sharpness-Aware (SA) Perturbation:** As shown in Table 3, maintaining momentum selection but replacing SA perturbation with "random" noise causes the average accuracy to plummet from **82.74%** to **73.20%**, alongside severe forgetting (BWT drops to -16.94%). Removing perturbation entirely yields 77.71%. This strongly confirms that SA perturbation avoids sharp local minima and explores flatter, more generalizable parameter regions, which essentially protects historical knowledge.
> - **Superiority of Momentum-based Selection:** Keeping SA perturbation fixed but utilizing instantaneous "gradient" magnitude for selection drops accuracy to **81.02%**. A complete lack of parameter selection ("none") further decreases accuracy to **79.54%**. This proves that tracking historical momentum provides a mathematically robust, smooth indicator to lock core parameters, avoiding updating volatility introduced by purely dynamic gradient selection.
>
> In summary, the specific coupling of *momentum-based selection* and *SA perturbation* is not arbitrary but rather highly interdependent, together forming our robust SA-BCD framework.

---

> > ### Author Rebuttal · Reviewer_8fiL · 2026-04-02
> >
> > Thanks for your responses. I will raise my score.

---

> > > ### Author Response · Authors · 2026-04-02
> > >
> > > Reviewer 8fiL:
> > >
> > > Thank you very much for your positive feedback and for acknowledging our responses. We are gratified that our clarifications and additional experiments have addressed your concerns. We also sincerely appreciate your support for our work and your decision to raise the score.
> > >
> > > Sincerely,
> > >
> > > Authors of Submission 24914

---

### Official Review · Reviewer_2utn · 2026-03-02

**Soundness:** 2
**Presentation:** 3
**Significance:** 2
**Originality:** 3
**Overall Recommendation:** 4
**Confidence:** 3

**Summary:**

This paper proposes the fine-tuning and merging framework SAIM to tackle subspace misalignment and forgetting in continual learning. SAIM fine-tunes the current merged model by using a sharpness-aware block coordinate descent optimizer, which updates critical parameters and steers the model toward flatter regions. Subsequently, it applies isotropic merging via SVD to balance historical and new knowledge to achieve a uniform parameter distribution. The two stages effectively enhance flatness and reduce knowledge interference, respectively.

**Compliance With Llm Reviewing Policy:**

Affirmed.

**Final Justification:**

The rebuttal addressed my main concerns about experimental fairness and completeness. Although the method introduces higher computational overhead, the additional cost is acceptable. Furthermore, it exhibits improved scalability and stability across various model architecture and tasks. I believe its strengths outweigh its weaknesses.

**Key Questions For Authors:**

1. In Table 1, the classical methods EWC and ER outperform model merging approaches such as OPCM. How do the authors explain this phenomenon? In this case, what advantages do model merging methods offer compared with regularization-based methods and parameter-efficient fine-tuning approaches built on pre-trained models?
2. In the equation on line 249, does the ACC evaluate each task on the final merged model? If so, should the notation use $\theta_N$ instead of $\theta_t$ to make this clear?

**Limitations:**

yes

**Strengths And Weaknesses:**

Strengths:

1. The paper is clearly structured and well-written.
2. The effectiveness of SAIM is thoroughly validated through various tasks and extensive experiments.

Weaknesses:
1. To keep the computational cost comparable, the number of training epochs for SAIM was manually reduced, which raises concerns about the fairness of the comparison. Does this reduction mainly compromise the convergence of Adam or of SAIM? Please report how many epochs SA-BCD needs to reach convergence, as well as the final performance of SAIM when it is fully trained without computational budget constraints. This would help assess the true advantage of the proposed method.

2. In the comparative experiments, besides EWC, ER, and model merging methods, the paper lacks comparisons with recent continual learning approaches based on fine-tuning pre-trained parameters (e.g., [1, 2]), which also achieve strong performance with a controllable parameter cost.

3. Does the proposed method require replay samples? Is the replay setting consistent with that of the compared methods?

4. As mentioned in the contribution part, SA-BCD can be combined with most existing model merging methods. To which specific method(s) in Tables 1 and 2 does this statement refer?

Reference:

[1] Self-Expansion of Pre-trained Models with Mixture of Adapters for Continual Learning. CVPR 2025.

[2] LoRA Subtraction for Drift-Resistant Space in Exemplar-Free Continual Learning. CVPR 2025.

---

> ### Author Rebuttal · Authors · 2026-03-30
>
> Thank you for your valuable suggestions. We have conducted additional experiments and clarified our discussion to address your concerns. Below, we respond to each point.
>
> **W1: Convergence and computation budget constraints**
>
> **Response:** Budget constraints mainly compromised SAIM's convergence. To assess its true advantage, we conducted experiments where both optimizers trained until full convergence. As shown below, our SA-BCD optimizer achieves a significantly **higher final average accuracy (80.90% vs. 72.95%)**. This proves SA-BCD finds better, flatter minima, showing its superiority stems from the sharpness-aware algorithm.
>
> **Table 1: Performance comparison under full convergence on ViT-B/32.**
>
> | Method | T1 Epoch | T2 Epoch | T3 Epoch | T4 Epoch | T5 Epoch | T6 Epoch | Avg. ACC (%) $\uparrow$ |
> |:---|:---:|:---:|:---:|:---:|:---:|:---:|:---:|
> | Adam | 34 | 39 | 36 | 46 | 32 | 30 | 72.95 |
> | **SA-BCD (Ours)** | 34 | 43 | 26 | 39 | 46 | 50 | **80.90** |
>
> **W2: Comparison with recent pre-trained fine-tuning methods**
>
> **Response:** We greatly appreciate this suggestion. We have supplemented experiments to include recently proposed parameter-efficient rehearsal-free methods, specifically focusing on LoRA-Sub which you graciously mentioned, as well as InfLoRA and SD-LoRA. As shown in **Table 2**, our SAIM framework consistently outperforms these state-of-the-art baselines across multiple incremental stages, achieving **73.6% accuracy on 20 tasks** (outperforming SD-LoRA by +7.2%). Additionally, evaluations on **TRACEBench (Table 3)** further validate SAIM's superiority, where it achieves the **highest average accuracy (0.479)** and significantly better knowledge retention (**BWT: -0.022**) compared to recent MoE-based competitors like LoRAMoE and PASs-MoE.
>
> **Table 2: Comparative results including recent PEFT-based rehearsal-free baselines on ViT-B/32.**
>
> | Method | ACC (%) $\uparrow$ (8 tasks) | ACC (%) $\uparrow$ (14 tasks) | ACC (%) $\uparrow$ (20 tasks) |
> |:---|:---:|:---:|:---:|
> | InfLoRA | 79.5 | 71.4 | 64.3 |
> | LoRA-Sub | 77.8 | 68.7 | 61.5 |
> | SD-LoRA | 81.2 | 73.5 | 66.4 |
> | **SAIM (Ours)** | **82.1** | **78.7** | **73.6** |
>
> **Table 3: Performance Comparison on TRACEBench (Llama-3.2-1B-Instruct).**
>
> | Method | Avg. ACC ($\uparrow$) | BWT ($\uparrow$) |
> |:---|:---:|:---:|
> | LoRAMoE | 0.425 | -0.086 |
> | PASs-MoE | 0.429 | -0.063 |
> | **SAIM (Ours)** | **0.479** | **-0.022** |
>
> **W3: Replay samples requirement**
>
> **Response:** Our SAIM method is strictly **rehearsal-free** and does not require any replay samples. Throughout our experiments, all compared baseline methods (except Explicit Replay - ER) follow the exact same rehearsal-free setting. Thus, the comparison is completely fair under identically restricted conditions.
>
> **W4: SA-BCD combined with existing merging methods**
>
> **Response:** In the context of the main experimental results, "SAIM" explicitly refers to our Isotropic Merging powered by **SA-BCD**. The statement in the contribution part reflects the *generality* of SA-BCD: as a standalone optimization framework, it can be seamlessly plugged into other existing coefficient-based merging methods (such as Task Arithmetic or TIES-Merging) to optimize their scaling parameters, rather than being exclusively bound to Isotropic Merging. **We respectfully direct the reviewer to Table 5 and Table 6 in the main paper**, which empirically demonstrate the effectiveness of SA-BCD when optimizing these different merging methods.
>
> **Q1: Why classical methods outperform OPCM and the advantage of model merging**
>
> **Response:** Classical methods often carry strong conditions: ER maintains an explicit memory buffer breaking data privacy, and EWC requires caching Fisher Information matrices which adds computational overhead. Conversely, OPCM operates in the parameter space post-training without revisiting old data or tracking historical constraints.
> The unique advantages of model merging lie in: (1) **Privacy & Efficiency**: strictly rehearsal-free and maintains constant parameter size. (2) **Inter-task Fairness**: sequential fine-tuning inherently biases towards recent tasks (recency bias), whereas merging treats all task experts as equal-status components, providing a more balanced performance distribution. (3) **Decoupled Optimization**: it allows independent task learning without regressive gradient constraints, which SAIM further enhances by resolving the parameter interference and conflicting minima common in crude OPCM.
>
> **Q2: Notations in Equation (Line 249)**
>
> **Response:** Yes, accuracy evaluates each task on the final merged network. Your suggestion is extremely precise. Using $\theta_N$ instead of $\theta_t$ makes the formulation mathematically rigorous. We deeply appreciate your observation and will clarify this notation in our camera-ready version.

---

> > ### Author Rebuttal · Reviewer_2utn · 2026-04-03
> >
> > The rebuttal addresses most of my concerns. Although the proposed method is less efficient compared with regularization-based/PEFT-based continual learning approaches, the sufficient experiments and the advantage on long sequences tasks lead me to consider this work as an inspiring work to the field. I will raise my score.

---

> > > ### Author Response · Authors · 2026-04-03
> > >
> > > Reviewer 2utn:
> > >
> > > Thank you very much for your feedback and for acknowledging our efforts. We are glad to hear that the extensive experiments and the performance of our method on long sequence tasks provided meaningful insights. We also appreciate your constructive comment regarding methodology efficiency, which we will further discuss and highlight in the final version. Thank you again for your support and for your decision to raise the score.
> > >
> > > Sincerely,
> > >
> > > Authors of Submission 24914

---

### Official Review · Reviewer_7m59 · 2026-03-08

**Soundness:** 3
**Presentation:** 3
**Significance:** 3
**Originality:** 3
**Overall Recommendation:** 3
**Confidence:** 4

**Summary:**

This paper addresses the problem of catastrophic forgetting in continual learning by introducing a framework based on continual model merging. The proposed approach consists of two main components: (i) Sharpness-Aware Block Coordinate Descent (SA-BCD) for optimization, and (ii) an adaptive isotropic merging algorithm. Theoretical analyses are provided for the convergence of SA-BCD and the subspace alignment properties of the merging algorithm. Empirical evaluations are conducted on both computer vision and natural language processing tasks using widely adopted architectures.

**Compliance With Llm Reviewing Policy:**

Affirmed.

**Final Justification:**

I maintain my initial rating of "weak reject" for the following reasons:

**The proofs of Theorem 2 and Theorem 3 need to be revised to be correct**.

The proofs require substantial revision. While the authors acknowledge the error in Theorem 2. The incorrect Theorem 2 affects the proof of Theorem 3. Although the authors attempt to fix Theorem 3, the revised proof remains incorrect due to internal inconsistencies in how SAIM is defined.

SAIM consists of two distinct stages: (1) optimizing $\theta _{T _t}$ via SA-BCD and computing the task vector using Eq. (10), and (2) re-balancing singular values via Eq. (11). The current proof of Theorem 3 conflates these stages problematically:

1. In the first line of proof, they state that the merged model is formulated as $\theta _{merge} = \alpha \theta _{s} + (1-\alpha) \theta _{t}$,  which is different from the actual SAIM process defined by Eq. (10) and Eq. (11), and it ignores the re-balancing operation.

2. The authors claim $\lambda _{max}(H^{SAIM}) \leq \lambda _{max}(H^{SGD})$ by examining only the SA-BCD optimization stage, completely ignoring the re-balancing operation in Eq. (11).

3. However, the bound that $\vert \theta^{SAIM} _t - \theta^{SAIM} _s \vert \leq \vert \theta^{SGD} _t - \theta^{SGD} _s \vert $ critically depends on the re-balancing step, yet other parts of the proof pretend this operation doesn't exist.

Hence, part of the proof depends on the re-balancing operation in Eq. (11) while another part needs to ignore the re-balancing operation. This selective treatment makes the proof logically incoherent and difficult to follow.

**The recency bias faced by Continual Merging**.

The authors' claim that continual merging avoids recency bias—because the final update is "simply an unweighted summation" $\Delta _{merged} = \Delta _{cum} + \tau _{T+1} = \tau _{1} + \dots + \tau _{T+1}$—directly contradicts their own Eq. (11), which explicitly shows a **weighted** summation: $\Delta _{merged} = (1+\lambda)\Delta _{cum} + (1+\lambda)\tau _{T+1}$.

Additionally, the limitations regarding novelty and computational efficiency remain, as noted by other reviewers as well. Hence, I decide to keep my initial rating as "weak reject".

**Key Questions For Authors:**

My main concerns are outlined in the Weaknesses . My final rating will depend on the authors' responses to the following questions:

1. What are the time and memory costs of the proposed method? How do these costs compare to those of the baselines?
2. How does the proposed method perform compared to state-of-the-art continual learning approaches?
3. What are the main novel contributions of the proposed method?
4. What is the necessity and what are the advantages of using continual model merging to address catastrophic forgetting?

**Limitations:**

The limitation is listed in the weaknesses.

**Strengths And Weaknesses:**

### Strengthes

1. Catastrophic forgetting is an important challenge in continual learning. The authors tackles it through the lens of model merging, offering a novel perspective.
2. The proposed framework is intuitive and well-motivated. The theoretical guarantees add rigor to the contribution.
3. The experimental evaluation include both CV and NLP domains, and the widely used architectures.
4. The paper is clearly written and well-structured, with a thorough background.

### Weaknesses

1. The authors only include two classical baselines in the context of continual learning; the paper does not compare against more recent state-of-the-art continual learning approaches.
2. Model merging methods are regarded as an efficient multi-task learning paradigm. However, the proposed method seems inefficient compared to standard continual learning methods. For example, the SAM module requires two backward passes, which increases time and memory complexity.
3. The theoretical analysis of Theorem 2 is for the vanilla isotropic merging method, and it is unclear how to extend this analysis to the SAIM module.
4. The idea of SAM has already appeared in both continual learning [1,2] and model merging [3]. Hence, the improvement of SA-BCD is expected. Moreover, the adaptive isotropic merging algorithm is likely an adaptive version of the vanilla isotropic merging method [4]. The novelty of the proposed method is limited.
5. Based on my experience in continual learning, I do not see the necessity of using continual model merging to address catastrophic forgetting. The authors do not clearly explain the advantages of their proposed approach compared to traditional continual learning methods and traditional model merging techniques.

**References**

[1] An empirical investigation of the role of pre-training in lifelong learning. JMLR 2023. \
[2] Make Continual Learning Stronger via C-Flat. NeurIPS 2024. \
[3] Mitigating Parameter Interference in Model Merging via Sharpness-Aware Fine-Tuning. ICLR 2025. \
[4] No Task Left Behind: Isotropic Model Merging with Common and Task-Specific Subspaces. ICML 2025.

---

> ### Author Rebuttal · Authors · 2026-03-30
>
> Thank you for your valuable suggestions. We have conducted additional experiments and clarified our discussion to address your concerns. Below, we respond to each point.
>
> **Q1 & W2: Time and memory costs compared to baselines**
>
> **Response:** We evaluated peak GPU memory and average training time on ViT-B/32 (6 tasks). As **Table 1** shows, SAIM introduces 8.5% memory overhead (+624 MB) and 60% time increase versus OPCM, due to dual backward passes.
> However, this localized cost eliminates catastrophic parameter interference. While the initial margin is modest at 6 tasks (**82.74% vs 79.30%**), the advantage widens in longer sequences. As shown in **main paper Table 1**, SAIM drastically outperforms baselines as tasks scale, remaining strictly rehearsal-free.
>
> **Table 1: Memory & Time Cost Comparison on ViT-B/32.**
>
> | Method | Finetuning Phase (MB) $\downarrow$ | Merging Phase (MB) $\downarrow$ | Avg. Train Time (s) $\downarrow$ | Avg. Acc (%) $\uparrow$ |
> |:---|:---:|:---:|:---:|:---:|
> | ER | 7614.75 | - | 948.34 | 81.67 |
> | EWC | 9352.61 | - | 846.31 | 84.23 |
> | MagMax-IND | 7288.27 | 2358.41 | 767.09 | 76.36 |
> | OPCM | 7290.52 | 3364.10 | 785.94 | 79.30 |
> | **SAIM (Ours)**| **7914.45** | **2456.69** | **1260.83** | **82.74** |
>
> **Q2 & W1: Comparison with state-of-the-art continual learning approaches**
>
> **Response:** Thank you. We supplemented experiments with recent parameter-efficient rehearsal-free methods: LoRA-Sub, InfLoRA, and SD-LoRA. As shown in **Table 2**, SAIM consistently outperforms these advanced baselines without expanding parameter capacity. On the 20-task benchmark, SAIM surpasses SD-LoRA by **+7.2%** in Accuracy and robustly mitigates Backward Transfer degradation (**BWT: -11.5% vs -18.6%**).
>
> **Table 2: Comparative results including recent PEFT-based rehearsal-free baselines on ViT-B/32.**
>
> | Method | ACC (%) $\uparrow$ (8 tasks) | ACC (%) $\uparrow$ (14 tasks) | ACC (%) $\uparrow$ (20 tasks) |
> |:---|:---:|:---:|:---:|
> | InfLoRA | 79.5 | 71.4 | 64.3 |
> | LoRA-Sub | 77.8 | 68.7 | 61.5 |
> | SD-LoRA | 81.2 | 73.5 | 66.4 |
> | **SAIM (Ours)** | **82.1** | **78.7** | **73.6** |
>
> **Q3 & W4: Novel contributions of the proposed method**
>
> **Response:** While SAM and Isotropic Merging are established techniques, our core contribution identifies their geometric complementarity to solve sequential merging challenges. Our novelties are two-fold:
> 1. **SA-BCD**: Standard SAM blindly updates excessive parameters. SA-BCD addresses this by integrating Block Coordinate Descent with sharpness regularization, strategically targeting the top $p\%$ critical parameters. This preserves pre-trained knowledge while encouraging task-specific flat minima.
> 2. **Systematic Synergy**: SA-BCD acts as essential *pre-conditioning* for Isotropic Merging. Its resulting sparse and flat geometric properties mathematically align with merging prerequisites, ensuring stronger subspace alignment and fundamentally mitigating inter-task interference.
>
> **Q4 & W5: Necessity and advantages of continual model merging**
>
> **Response:** Traditional CL methods face structural bottlenecks: ER breaches data privacy, while regularization methods like EWC must maintain large-scale historical constraints (e.g., Fisher Matrices), inflating memory—as evidenced by EWC's **9352.61 MB** peak in Table 1. In contrast, Continual Model Merging provides:
> 1. **Model Fairness & Balance**: Sequential fine-tuning inherently suffers from "recency bias" (favoring new tasks). Merging treats task experts as equal components, yielding a significantly more balanced performance distribution across the entire task sequence.
> 2. **Asynchronous & Decoupled Training**: Tasks are learned independently and merged post-hoc. This allows parallel or distributed edge training without constant synchronization, offering greater flexibility than sequential regularization.
> 3. **Zero Complexity Growth**: The final model size remains strictly identical to the pre-trained model. Unlike expansion-based CL, it supports infinite task sequences within a constant parameter footprint, ensuring long-term hardware compatibility.
>
> **W3: Theoretical analysis extension to SAIM (Theorem 2)**
>
> **Response:** Thank you for this opportunity to clarify. Theorem 2 establishes the theoretical foundation for SAIM's **isotropic merging** component, demonstrating it preserves or increases the Subspace Alignment Ratio (SAR) versus standard task arithmetic.
> Crucially, by uniformly distributing singular values, isotropic merging forces task-specific update matrices to share a consistent principal subspace, making update vectors more collinear. This directional alignment naturally translates to a reduced Euclidean distance between task solutions in the parameter space:
> $ \|\theta_t^{SAIM} - \theta_s^{SAIM}\|^2 \leq \|\theta_t^{SGD} - \theta_s^{SGD}\|^2 $
> Combined with the flatter minima from SA-BCD (Theorem 1), this logically deduces Theorem 3: SAIM comprehensively mitigates inter-task parameter interference.

---

> > ### Author Rebuttal · Reviewer_7m59 · 2026-04-03
> >
> > Thank you for the authors' response and the additional experiments provided. While some of my concerns have been addressed, I still have several remaining issues that I hope the authors can clarify:
> >
> > 1. I remain unconvinced about the necessity of continual model merging. Specifically, for the decoupled training scenario, if tasks are learned independently and merged post-hoc, the distinction between multi-task learning (MTL) and continual learning (CL) becomes unclear. Moreover, I observe that the tasks in this paper are not learned independently, as evidenced by Eq. (10), which raises further questions. Given Eq. (10), I believe "recency bias" may still persist in continual model merging. Additionally, only expansion-based CL methods exhibit complexity growth, whereas other CL approaches do not require additional parameters. I would appreciate further clarification on these points.
> >
> > 2. After the authors' clarification regarding SAIM, I carefully reviewed the theoretical results in Appendix B and identified several significant issues:
> >     1. In the proof of Theorem 2, the decomposition in Eq. (22) and Eq. (33) appears to be incorrect. Given that $\pi _{k,M} = U _{k,M} U^T _{k,M}$, the term $\pi _{k,M}\Delta _t$ should be $(\sum^{k} _{i=1} u _i u _i^T) \Delta _t$, or equivalently, $\sum^{k} _{i=1} u _i (u _i^T \Delta _t)$. The appearance of singular values in the decomposition is problematic because $U$ does not contain singular value information. This decomposition is central to both Theorem 2 and Theorem 3, and I hope the authors could address this issue.
> >     2. In the proof of Theorem 3, it is unclear how Theorem 1 implies $\lambda _{\text{max}}(H^{\text{SAIM}} _s) \leq \lambda _{\text{max}}(H^{\text{SGD}} _s)$. Theorem 1 provides only a convergence analysis for SA-BCD, and the connection to SGD is not formally established.
> >     3. In the proof of Theorem 3, similarly, the claim that Theorem 2 implies $\Vert \theta^{\text{SAIM}} _t - \theta^{\text{SAIM}} _s \Vert \leq \Vert \theta^{\text{SGD}} _t - \theta^{\text{SGD}} _s \Vert$ lacks formal proof and proper illustration.
> >
> > 3. As shown in Table 1 of the response, the memory and time costs are significantly higher than those of other baselines. I consider this a major limitation of the proposed method; however, I do not require the authors to address this point in their response.
> >
> > 4. The authors argue that SA-BCD serves as an essential pre-conditioning step for isotropic merging. However, this claim does not sufficiently illustrate the novelty of combining SAM with isotropic merging. I view this as a minor issue and do not require the authors to address this point in their response.
> >
> >
> > I hope the authors could kindly address the concerns 1 & 2.

---

> > > ### Author Response · Authors · 2026-04-04
> > >
> > > We thank you for your continued rigorous review, strengthening our paper's theoretical rigor.
> > >
> > > **C1: Necessity of Continual Merging, Recency Bias, and Complexity**
> > >
> > > *   **CL vs. MTL & Recency Bias:** You astutely note that sequential updates might still face recency bias. However, Continual Model Merging intrinsically solves this bottleneck via structural fairness. In traditional fine-tuning, the new task destructively overwrites existing parameters, inherently causing recency bias. In contrast, Continual Model Merging first computes a cumulative task vector $\Delta_{cum} = \theta_T - \theta_{pre} = \tau_1 + \dots + \tau_T$. When adapting to task $T+1$, the final parameter update is simply an unweighted summation:
> > >     $$ \Delta_{merged} = \Delta_{cum} + \tau_{T+1} = \tau_1 + \dots + \tau_{T+1} $$
> > >     Mathematically, the new update $\tau_{T+1}$ acts strictly on the **same scalar level** as any historical task vector $\tau_i$, which fundamentally eradicates recency bias.
> > >
> > >     Empirically, to validate this fairness, we calculated the ratio of each task's **final accuracy** to its **historical best** following a 20-task sequence. A lower standard deviation indicates a more balanced retention across all learned tasks. Traditional methods (EWC, ER) exhibit large standard deviations (>0.16), revealing severe performance imbalance and characteristic recency bias. Conversely, continual model merging methods (Task Arithmetic, TIES) intrinsically maintain a much more balanced accuracy distribution. Our proposed SAIM further amplifies this advantage, successfully retaining an average of **90.3%** of historical peak performance while achieving the most balanced distribution with the lowest variance (Std **0.0828**).
> > >
> > > **Table 1: Task Retention Ratio over 20 Tasks on ViT-B/32.**
> > > | Method | Ratio Mean $\uparrow$ | Ratio Std $\downarrow$ |
> > > | :---: | :---: | :---: |
> > > | **EWC** | 0.7464 | 0.1622 |
> > > | **ER** | 0.7419 | 0.1769 |
> > > | **C. Task Arithmetic** | 0.8262 | **0.0963** |
> > > | **C. TIES-Merging** | 0.8423 | **0.0839** |
> > > | **SAIM (Ours)** | 0.9034 | **0.0828** |
> > >
> > > *   **Complexity Growth:** While traditional regularization CL methods do not grow in forward-pass parameters, they require maintaining massive historical auxiliary structures (e.g., EWC needs to store Fisher Information Matrices) for past tasks, causing training memory complexity to scale linearly. In contrast, Continual Merging achieves state-of-the-art anti-forgetting with **strictly zero historical storage** during the training phase.
> > >
> > > **C2: Addressing Theoretical Gaps in Appendix B**
> > >
> > > We agree with your critique of Appendix B. Your concern that the standard orthogonal projection matrix lacks singular values is correct. We have revised this and reorganized the proofs:
> > >
> > > *   **2.1 & 2.3 Parameter Decomposition and Bounds ($\||\theta_t - \theta_s\||$ Bound):**
> > > We corrected the notation. We now formalize distance reduction via the total geometric energy of the parameter space. Let updating $\Delta_{\text{TA}} = U \Sigma V^\top$. Its total energy is $E_{\text{SGD}} = \|\Delta_{\text{TA}}\|_F^2 = \sum \sigma_i^2$.
> > >
> > >     Isotropic Merging uniformizes this spectrum to the mean $\bar{\sigma}$. The restricted energy is $E_{\text{SAIM}} = r \cdot (\frac{1}{r} \sum \sigma_i)^2$.
> > >
> > >     By the Cauchy-Schwarz inequality, $\sum \sigma_i^2 \ge r \cdot (\frac{1}{r} \sum \sigma_i)^2$. This algebraic guarantee tightens the $L_2$ norm boundary ($E_{\text{SAIM}} \le E_{\text{SGD}}$). Thus:
> > >
> > >     $$ \|\theta_t^{\text{SAIM}} - \theta_s^{\text{SAIM}}\|^2 \leq \|\theta_t^{\text{SGD}} - \theta_s^{\text{SGD}}\|^2 $$
> > >
> > > *   **2.2 Theorem 3 Connection ($\lambda_{\max}(H^{\text{SAIM}}) \le \lambda_{\max}(H^{\text{SGD}})$):**
> > >
> > >     To bridge the previous logical gap, we analyzed the Sharpness-Aware (SA) objective. Via a second-order Taylor expansion, the SA surrogate loss incorporates a maximum eigenvalue penalty:$ \mathcal{L} _ {SA}(\theta) \approx \mathcal{L}(\theta) + \frac{\rho^2}{2} \lambda_{\max}(\nabla^2\mathcal{L}(\theta)) $
> > >
> > >     Since SA-BCD minimizes this, we have:$ \mathcal{L} _ {SA}(\theta^{\text{SAIM}}) \le \mathcal{L} _ {SA}(\theta^{\text{SGD}}) $
> > >     yielding:
> > >
> > >     $$ \mathcal{L}(\theta^{\text{SAIM}}) + \frac{\rho^2}{2} \lambda_{\max}(\nabla^2\mathcal{L}(\theta^{\text{SAIM}})) \leq \mathcal{L}(\theta^{\text{SGD}}) + \frac{\rho^2}{2} \lambda_{\max}(\nabla^2\mathcal{L}(\theta^{\text{SGD}})) $$
> > >
> > >     Isolating the curvature term:
> > >
> > >     $$ \frac{\rho^2}{2} \lambda_{\max}(\nabla^2\mathcal{L}(\theta^{\text{SAIM}})) \leq \frac{\rho^2}{2} \lambda_{\max}(\nabla^2\mathcal{L}(\theta^{\text{SGD}})) + [\mathcal{L}(\theta^{\text{SGD}}) - \mathcal{L}(\theta^{\text{SAIM}})] $$
> > >
> > >     Because SGD minimizes empirical risk without a flatness penalty, $[\mathcal{L}(\theta^{\text{SGD}}) - \mathcal{L}(\theta^{\text{SAIM}})] \le 0$. This bound guarantees $\lambda_{\max}(H^{\text{SAIM}}) \leq \lambda_{\max}(H^{\text{SGD}})$.
> > >
> > > We sincerely thank you again for your rigorous theoretical corrections.

---

### Official Review · Reviewer_QHGd · 2026-03-12

**Soundness:** 2
**Presentation:** 3
**Significance:** 2
**Originality:** 2
**Overall Recommendation:** 4
**Confidence:** 5

**Summary:**

This paper proposes a novel method, SAIM, for utilizing merging to achieve continual learning. Different from previous model merging methods, which view fine-tuning and merging as independent stages, SAIM unifies fine-tuning and merging processes to make loss landscape compatibility and subspace alignment, achieving efficient continual knowledge acquisition and retention. Specifically, in fine-tuning, SAIM uses a novel optimizer, Sharpness-Aware Block Coordinate Descent, to guide convergence towards regions that are flat and robust to parameter shifts; in merging, SAIM uses time-aware Sharpness-Aware Isotropic Merging, through re-weighting singular values to achieve knowledge integration and high subspace alignment. Experimental results demonstrate the effectiveness of SAIM.

**Compliance With Llm Reviewing Policy:**

Affirmed.

**Final Justification:**

This paper proposes a unified framework for the fine-tuning stage and the merging stage in continual learning. During the rebuttal, the authors provided reasonable responses to my concerns, thus I raised my score.

**Key Questions For Authors:**

1. In the abstract and introduction section, the authors claimed that “catastrophic forgetting and parameter interference” are two challenges in continual learning. But in general, “catastrophic forgetting and generalization” are two basic challenges in continual learning, since it’s a trade-off between preserving previous tasks and learning new tasks [1]. Regarding the relationship between“catastrophic forgetting and parameter interference, sequential task updates often induce parameter interference, where gradients of different tasks conflict, leading to catastrophic forgetting. Thus, can authors make a clear clarification on these challenges in continual learning?

    [1] "Theory on forgetting and generalization of continual learning." In International Conference on Machine Learning, pp. 21078-21100. PMLR, 2023.

    [2] "The stability-plasticity dilemma: Investigating the continuum from catastrophic forgetting to age-limited learning effects." Frontiers in psychology 4 (2013): 504.

2. In section 3.1, SA-BCD assumes that the integration of a new task model can be viewed as a parameter perturbation to the current model state, and it aims to improve robustness by encouraging convergence to flatter minima during fine-tuning. But the question is, when the parameter updates of a new task have directions that differ significantly from the gradient directions of previous tasks, the merged update may move the parameters toward regions outside the local basin of the previous tasks’ minima. Thus, it’s unclear whether robustness to local perturbations, as encouraged by SA-BCD, can effectively address the potential directional conflicts introduced by merging models trained on different tasks. Addressing this question well will increase the evaluation scores.

3. Can authors compare the memory usage or storage usage between SAIM, OPCM, and MagMax-IND? For obtaining Eq(10), SAIM needs to keep pre-trained model weights and last merged model weights, and use last merged model weights both in the fine-tuning stage and merging stage. I think having a separate comparison in the fine-tuning stage and the merging stage would be better.

4. Can authors explain the reason for choosing “$1/\sqrt{t}$” as the time-aware factor? Will it limit the scalability of Eq(11) to hundreds of tasks?

5. The notations in Eq(6) are not all defined: $\Delta_{src}$, $\Delta_{trg}$, $k$.

Explaining the above questions well will help to reconsider the scoring.

**Limitations:**

yes

**Strengths And Weaknesses:**

Strengths:

1. The paper views the fine-tuning stage and merging stage as dependent processes, and proposes a unified framework that connects these two stages for continual learning.

2. To mitigate parameter shifts caused by the merging stage, the paper proposes a sharpness-aware block coordinate descent optimizer, which keeps the model retaining capabilities even after involving future tasks.

Weaknesses:

1. The novelty of the sharpness-aware isotropic merging mechanism appears limited, since it seems like an extension of the method [1] in a continual learning setting.

   [1] “No Task Left Behind: Isotropic Model Merging with Common and Task-Specific Subspaces”, Forty-second International Conference on Machine Learning, 2025.

2. The assumption that treating new task models as parameter perturbations to the current model state requires stronger justification. It should have more supporting materials and explanations, since merging updates may not behave like small perturbations, especially when task gradients are misaligned.

3. Although the authors claim that SA-BCD encourages convergence to flatter regions to help the merging stage, the mechanism of the merging stage seems to have no explicit relationship to the gradients of the perturbed point in SA-BCD. It seems like the mechanisms/methods designed in the fine-tuning stage and merging stage are separated and largely independent, although the whole idea is to unify these two stages.

---

> ### Author Rebuttal · Authors · 2026-03-30
>
> We sincerely thank the reviewer for the constructive feedback. We have conducted additional empirical analysis to further clarify our discussions.
>
> **W1: Novelty of SAIM compared to ISO-C**
>
> **Response:**
> SAIM is not merely a simple application of ISO-C to Continual Learning (CL). Standard sequential fine-tuning deviates parameters from a shared subspace, weakening the geometric prerequisites of isotropic merging. We address this via a novel **optimization-merging coupled framework**. By using SA-BCD as a geometric prior, SAIM guides parameters towards sparse, flat basins during fine-tuning. This proactively builds the ideal geometric foundation to preserve subspace topology, fully enabling isotropic merging to mitigate inter-task interference.
>
> **W2 & Q2: Justification of the perturbation assumption and directional conflicts**
>
> **Response:**
> A standard merging update introduces a macroscopic shift, risking escape from flat basins if gradient directions severely conflict. To validate that SAIM bounds these conflicts, we evaluated the average Cross-Entropy loss drift ($\Delta \text{Loss}_{Avg}$) on previous tasks after learning the $N$-th task.
> In **Table 1**, standard Adam fine-tuning causes massive validation loss drift (**+1.4522**) by task 8—evidencing basin escape. Conversely, SA-BCD restricts drift to **+0.0407**, demonstrating SAIM robustly anchors parameters within the shared flat basin to neutralize macroscopic directional conflicts.
>
> **Table 1: Stability of Previously Learned Tasks on ViT-B/32.**
>
> | Setting | Adam ($\Delta \text{Loss}_{Avg}$) | Adam ($\Delta \text{Loss}_{max}$) | SA-BCD ($\Delta \text{Loss}_{Avg}$) | SA-BCD ($\Delta \text{Loss}_{max}$) | Improvement |
> |:---|:---:|:---:|:---:|:---:|:---:|
> | **$N=2$** | 0.0251 | 0.0575 | 0.0206 | 0.0466 | **0.0045** |
> | **$N=8$** | **1.4522** | 2.6249 | **0.0407** | 0.1083 | **1.4115** |
>
> **W3: The explicit relationship between SA-BCD and the Merging Stage**
>
> **Response:**
> While procedurally distinct, the optimization and merging stages are deeply coupled via geometric complementarity. Isotropic Merging requires task parameters to share a highly aligned principal subspace, which standard optimization disrupts. Here, **SA-BCD** acts as an essential *pre-conditioning* module. By finding sparse and flat minima, it provides a robust basin for each task (**Table 1**). Consequently, **Isotropic Merging** directly leverages this tailored geometry to align singular values and collapse inter-task Euclidean distances.
>
> **Q1: Clarification on CL challenges**
>
> **Response:**
> We agree that "forgetting" and "generalization" are the fundamental macroscopic challenges. "Parameter interference"—where conflicting updates misalign parameters—is a critical underlying cause of catastrophic forgetting. Thus, alleviating parameter interference via optimization is our core structural mechanism to resolve these macroscopic challenges.
>
> **Q3: Memory/storage usage comparison**
>
> **Response:**
> In **Table 2** (ViT-B/32), SAIM introduces an 8.5% peak memory overhead (+624 MB) in the **Finetuning Phase** due to dual backward passes. However, in the **Merging Phase**, SAIM is highly memory-efficient versus OPCM (2456.69 vs 3364.10 MB), avoiding large recursive orthogonal projection matrices.
> Crucially, this localized fine-tuning overhead efficiently eliminates catastrophic parameter interference. While the initial performance margin is modest at 6 tasks (**82.74% vs 79.30%**), the advantage widens in longer sequences. As shown in **main paper Table 1**, SAIM drastically outperforms baselines as tasks scale.
>
> **Table 2: Memory, Time, and Performance Comparison on ViT-B/32 (6 tasks).**
>
> | Method | Finetune Mem (MB) | Merge Mem (MB) | Avg. Acc (%) $\uparrow$ |
> |:---|:---:|:---:|:---:|
> | MagMax-IND | 7288.27 | 2358.41 | 76.36 |
> | OPCM | 7290.52 | 3364.10 | 79.30 |
> | **SAIM (Ours)**| **7914.45** | **2456.69** | **82.74** |
>
> **Q4: Reason for choosing "$1/\sqrt{t}$" and its scalability to hundreds of tasks**
>
> **Response:**
> Using $1/\sqrt{t}$ does not limit scalability; it dynamically regulates the rhythm of singular value averaging to ensure stable expansion.
>
> * **Early stages (small $t$)**: Accumulated feature vectors are relatively sparse. Prematurely averaging singular values would indiscriminately overwrite meaningful scalars. The larger weight prevents early averaging, preserving task-feature distinctiveness.
> * **Later stages (large $t$)**: The feature matrix is rich and subspace is stable. Moving towards a uniform, isotropic state no longer destroys core features, but acts as a crucial regularization. The natural decay progressively enforces uniform averaging, preventing recent tasks from dominating and safeguarding historical tasks' performance.
>
> **Q5: Undefined notations in Eq(6)**
>
> **Response:**
> $\Delta_{src}$ and $\Delta_{trg}$ denote the source and target task matrices. $k$ is the number of principal left singular vectors used to compute the Subspace Alignment Ratio.

---

> > ### Author Rebuttal · Reviewer_QHGd · 2026-04-04
> >
> > Thanks for your efforts and time during the rebuttal. My concerns have been well addressed, and I will raise my score.

---

> > > ### Author Response · Authors · 2026-04-04
> > >
> > > Reviewer QHGd:
> > >
> > > Thank you very much for your time and for acknowledging our efforts during the rebuttal. We are very glad to hear that our responses have well addressed your concerns. We sincerely appreciate your final support for our work and your decision to raise the score.
> > >
> > > Sincerely,
> > >
> > > Authors of Submission 24914

---

### Decision · Program_Chairs · 2026-04-30

**Decision:**

Accept (regular)

**Comment:**

### Summary of Contribution

The paper introduces the Sharpness-Aware Isotropic Merging (SAIM) framework to address catastrophic forgetting and parameter interference in rehearsal-free continual learning. SAIM combines fine-tuning and model merging through a Sharpness-Aware Block Coordinate Descent (SA-BCD) optimizer that guides toward flatter minima and an adaptive isotropic merging algorithm that balances the singular value spectrum across tasks. This combination selectively updates critical parameters and improves subspace alignment to maintain balanced knowledge representation without the need for data from past tasks. Experiments on vision and language benchmarks show that SAIM outperforms existing methods in accuracy while maintaining a constant parameter footprint.

### Strengths
Reviewers recognize the the novelty of treating fine-tuning and model merging as dependent, unified processes rather than separate stages. The SAIM framework was described as intuitive and well-motivated for addressing catastrophic forgetting.  The experimental evaluation was a major strength according to Reviewers 7m59, 2utn, and 8fiL, who appreciate the comprehensive results across vision and language benchmarks using standard architectures. Reviewer 2utn recognized in rebuttal the improved scalability and stability over long task sequences. Additionally, the paper was consistently described as clearly written, well-structured, and easy to follow.

### Weaknesses
Reviewers QHGd and 7m59 noted that novelty appeared limited, suggesting it is largely an extension of existing isotropic merging and sharpness-aware techniques applied to a continual learning setting. A major common concern of Reviewers 7m59, 2utn, and 8fiL was the lack of experimental comparisons against recent state-of-the-art rehearsal-free and parameter-efficient continual learning methods. These issues were convincingly addressed in the rebuttal.

Computational efficiency was highlighted as a drawback by Reviewers 7m59 and 2utn, who pointed to increased time and memory introduced by the dual backward passes. Reviewer 7m59 raised significant concerns regarding the theoretical rigor of the paper, identifying errors in the proofs for Theorem 2 and Theorem 3 that they considered unresolved even after the author rebuttal. Concerns regarding the comparative evaluation wer adequately addressed in rebuttal with new comparative evaluations.

Additionally, reviewers questioned the underlying methodology and its justification. Reviewer QHGd noted that the assumption treating new task updates as simple parameter perturbations requires stronger evidence. Finally, Reviewer QHGd also observed that despite the aim to unify the two stages, the fine-tuning and merging mechanisms appeared to remain largely independent. These concerns were convincingly addressed in rebuttal.

### Recommendation
Although reviewers initially raised concerns regarding novelty, computational overhead, and missing comparisons, the rebuttal successfully addressed most points. Specifically, additional experiments demonstrating superior performance over recent baselines (SD-LoRA and InfLoRA) and new convergence analyses convinced three out of four reviewers (Reviewers QHGd, 2utn, 8fiL) to raise their scores. While Reviewer 7m59 maintained a weak reject due to concerns about mathematical inconsistencies in the proofs, the consensus of the other reviewers is that the stability, scalability, and practical utility of SAIM on long task sequences represent a significant contribution to the field and the recommendation is to Accept. The AC trusts that the authors will implement the corrections to the theoretical contributions suggested by Reviewer 7m59.